**ShellChron 0.4.0: A new tool for constructing chronologies in accretionary carbonate archives**
**from stable oxygen isotope profiles**
Niels J. de Winter[1,2]
[1]Department of Earth Sciences, Utrecht University, Utrecht, the Netherlands
[2]AMGC research group, Vrije Universiteit Brussel, Brussels, Belgium
Corresponding author: Niels J de Winter (n.j.dewinter@uu.nl)
**Abstract**
This work presents ShellChron, a new model for generating accurate internal age models for high-
resolution paleoclimate archives, such as corals, mollusk shells and speleothems. Reliable sub-annual
age models form the backbone of high-resolution paleoclimate studies. In absence of independent sub-
annual growth markers in many of these archives, the most reliable method for determining the age of
samples is through age modelling based on stable oxygen isotope or other seasonally controlled proxy
records. ShellChron expands on previous solutions to the age model problem by fitting a combination
of a growth rate and temperature sinusoid to model seasonal variability in the proxy record in a sliding
window approach. This new approach creates smoother, more precise age-distance relationships for
multi-annual proxy records with the added benefit of allowing assessment of the uncertainty on the
modelled age. The modular script of ShellChron allows the model to be tailored to specific archives,
without being limited to oxygen isotope proxy records or carbonate archives, with high flexibility in
assigning the relationship between the input proxy and the seasonal cycle. The performance of
ShellChron in terms of accuracy and computation time is tested on a set of virtual seasonality records
and real coral, mollusk and speleothem archives. The result shows that several key improvements in
comparison to previous age model routines enhance the accuracy of ShellChron on multi-annual records
while limiting its processing time. The current full working version of ShellChron enables the user to
model the age of a 10-year long high-resolution (16 samples/yr) carbonate records with monthly
accuracy within one hour of computation time on a personal computer. The model is freely accessible
on the CRAN database and GitHub. Members of the community are invited to contribute by adapting
the model code to suit their research topics and encouraged to cite the original work of Judd et al. (2018)
alongside this work when using ShellChron in future studies.
**1. Introduction**
Fast growing carbonate archives, such as coral skeletons, mollusk shells and speleothems, contain a
wealth of information about past and present climate and environment (e.g. Urban et al., 2000; Wang et
al., 2001; Steuber et al., 2005; Butler et al., 2013). Recent advances in analytical techniques have
improved our ability to extract this information and obtain records of the conditions under which these
carbonates precipitated at high temporal resolutions, often beyond the annual scale (Treble et al., 2007;
Saenger et al., 2017; Vansteenberge et al., 2019; de Winter et al., 2020a; Ivany and Judd, 2022). Key
to the interpretation of such records is the development of reliable chemical or physical proxies for
climate and environmental conditions which can be measured on a sufficiently fine scale to allow
variability to be reconstructed at the desired time resolution. Examples of suitable proxies include
observations of variability in carbonate fabric and microstructure and in (trace) elemental and isotopic
composition (Frisia et al., 2000; Lough, 2010; Ullmann et al., 2010; Schöne et al., 2011; Ullmann et al.,
2013; Van Rampelbergh et al., 2014; de Winter et al., 2017). The unique preservation potential of
carbonates in comparison with archives of climate variability at similar time resolutions, such as tree ring
records and ice cores, now allows us to recover information about climate and environment of the
geological past from these proxies on the (sub-)seasonal scale (Ivany and Runnegar, 2010; Ullmann
and Korte, 2015; Vansteenberge et al., 2016; de Winter et al., 2018; 2020b; c; Mohr et al., 2020). The
importance of this development cannot be overstated because variability at high (daily and seasonal)
resolution constitutes the most significant component of climate variability (Mitchell, 1976; Huybers and
Curry, 2006; Zhu et al., 2019; von der Heydt et al., 2021). Accurate reconstructions of this type of
variability are therefore fundamental to our understanding of Earth's climate system and critical for
projecting its behavior in the future under anthropogenic global warming conditions (IPCC, 2021).

A reliable age model is crucial for the interpretation of high-resolution carbonate records. An age model is defined as a set of rules or markers that allows the translation of the location of a measurement or observation on the archive to the time at which the carbonate was precipitated. This translation is required for aligning records from multiple proxies or archives on a common time axis. Age alignment enables data to be intercomparable and to be interpreted in the context of processes playing a role at similar timescales. Age models are based on knowledge about the growth or accretion rate of the archive through time. Many high-resolution carbonate archives contain growth markers on which age models can be based (e.g. Jones, 1983; Le Tissier et al., 1994; Verheyden et al., 2006). These are especially valuable in some mollusk species, in which growth lines demarcate annual, daily, or even tidal cycles (e.g. *Arctica islandica*, Schöne et al., 2005; *Pecten maximus*, Chavaud et al., 2005 and *Cerastoderma edule*, Mahé et al., 2010). However, in many mollusk species and most carbonate archives, such independent growth indicators are absent or too infrequent to (relatively) date high-resolution measurements (Judd et al., 2018; Huyghe et al., 2019). In such cases, age models need to be based on alternative indicators.

The oxygen isotope composition of carbonates ($\delta^{18}O_c$) is closely dependent on the isotopic composition of the fluid ($\delta^{18}O_w$) and the temperature at which the carbonate is precipitated (Urey, 1948; McCrea, 1950; Epstein et al., 1953). In most natural surface environments, either one or both factors is strongly dependent on the seasonal cycle, one generally being dominant over the other. This causes carbonates precipitated in these environments to display strong quasi-sinusoidal variations in $\delta^{18}O_c$ that record the seasonal cycle (e.g. Dunbar and Wellington, 1981; Jones and Quitmyer, 1996; Baldini et al., 2008). Examples of this behavior include seasonal cyclicity in sea surface temperatures recorded in the $\delta^{18}O_c$ of corals and mollusks and seasonal cyclicity in the $\delta^{18}O_w$ of precipitation recorded in speleothems (Dunbar and Wellington, 1981; Schöne et al., 2005; Van Rampelbergh et al., 2014). This relationship is challenged in tropical latitudes, where temperature seasonality is restricted. However, in some tropical archives, the annual cycle of $\delta^{18}O_w$ in precipitation still allows the annual cycle to be resolved from $\delta^{18}O$ records (e.g. Evans and Schrag, 2004). These properties make $\delta^{18}O_c$ one of the most highly sought-after proxies for climate variability, and high-resolution $\delta^{18}O_c$ records are abundant in the paleoclimate literature (e.g. Lachniet, 2009; Lough, 2010; Schöne and Gillikin, 2013 and references therein).

The close relationship between $\delta^{18}O_c$ records and the seasonal cycle can also be exploited to estimate variability in growth rate of the archive. This property of $\delta^{18}O_c$ curves has been recognized by previous

authors, and attempts have been made to quantify intra-annual growth rates from the shape of $\delta^{18}O_c$
profiles (Wilkinson and Ivany, 2002; Goodwin et al., 2003; De Ridder et al., 2006; Goodwin et al., 2009;
De Brauwere et al., 2009; Müller et al., 2015; Judd et al., 2018). Over time, these so called "growth
models" have improved from fitting of sinusoids to $\delta^{18}O_c$ data (Wilkinson and Ivany, 2002; De Ridder et
al., 2006) to including increasingly complicated (inter)annual growth rate curves to the model to fit the
shape of the $\delta^{18}O_c$ data (Goodwin et al., 2003; 2009; Müller et al., 2015; Judd et al., 2018). These later
models manage to fit the shape of $\delta^{18}O_c$ records well, but they often rely on detailed *a priori* knowledge
of growth rate or temperature patterns (e.g. Goodwin et al., 2003; 2009), which requires measurements
of one or more parameters in the environment. These measurements are not available in studies on
carbonate archives from the archeological or geological past. In contrast, the latest model by Judd et al.
(2018; GRATAISS, or "Growth Rate and Temporal Alignment of Isotopic Serial Samples") is based only
on the assumption that growth and temperature follow quasi-sinusoidal patterns and can therefore work
with $\delta^{18}O_c$ data alone, making it more widely applicable. The simplified parameterization of temperature
and growth rate seasonality by Judd et al. (2018) using two (skewed) sinusoids is demonstrated to
approximate natural circumstances very well.
However, the GRATAISS model is still limited in its use because it requires whole, individual growth
years to be analyzed separately, resulting in a discontinuous time series when applied on records
containing multiple years of $\delta^{18}O_c$ data and no solution for incomplete years. In addition, the model has
no option to supply information about the less dominant factor that drives $\delta^{18}O_c$ values ($\delta^{18}O_w$ of sea
water in the case of mollusks and corals). Furthermore, only estimates from aragonite records are
supported, while the $\delta^{18}O_c$ value of the other dominant carbonate mineral, calcite, has a different
temperature relationship (Kim and O'Neil, 1997). Finally, neither of the models highlighted above except
for the MoGroFun model by Goodwin et al. (2009) include any assessment of the uncertainty of the
constructed age model.
Here, a new model for estimating ages of samples in seasonal $\delta^{18}O_c$ curves is presented which
combines the advantages of previous models while attempting to negate their disadvantages.
ShellChron combines a skewed growth rate sinusoid with a sinusoidal temperature curve to model $\delta^{18}O_c$
using the Shuffled Complex Evolution model developed at the University of Arizona (SCEUA; Duan et
al., 1992; following Judd et al., 2018). It applies this optimization using a sliding window through the
dataset (as in Wilkinson and Ivany, 2002) and includes the option to use a Monte Carlo simulation
approach to combine uncertainties on the input ($\delta^{18}O_c$ and sample distance measurements) and the
model routine (as in Goodwin et al., 2009). As a result, ShellChron produces a continuous time series
with a confidence envelope, supports records from multiple carbonate minerals and allows the user to
provide information on the less dominant variable influencing $\delta^{18}O_c$ (e.g. $\delta^{18}O_w$) if available (see **section**
**2**). The modular design of ShellChron's functional script allows parts of the model to be adapted and
interchanged, supporting a wide range of climate and environmental archives. As a result, the initial
design of ShellChron for reconstructing age models in temperature-dominated $\delta^{18}O_c$ records from
marine bio-archives (e.g. corals and mollusks) presented here can be easily modified for application on
other types of records. The routine is worked out into a ready-to-use package for the open-source
computational programming language R and is directly available without restrictions, allowing all
interested parties to freely modify and build on the base structure to adapt it to their needs (R Core
Team, 2020; full package code and documentation in **SI1**, see also **Code availability**).

**2. Scientific basis**
The relationship between $\delta^{18}O_c$ and the temperature of carbonate precipitation was first established by
Urey (1951) and later refined with additional measurements and theoretical models (e.g. Epstein et al.,
1953; Tarutani et al., 1969; Grossman and Ku, 1986; Kim and O'Neil, 1997; Coplen, 2007; Watkins et
al., 2014; Daëron et al., 2019). Empirical transfer functions for aragonite and calcite by Grossmann and
Ku (1986; modified by Dettmann et al., 1999; **equation 1**) and Kim and O'Neil (1997; **equation 2**, with
VSMOW to VPDB scale conversion following Brand et al., 2014; **equation 3**) have so far found most
frequent use in modern paleoclimate studies and are therefore applied as default relationships in the
ShellChron model (see *d18O_model* function).
$$T[°C] = 20.6 - 4.34 * (\delta^{18}O_c[‰VPDB] - \delta^{18}O_w[‰VSMOW] + 0.2) \ (\mathbf{1})$$
$$1000 * \ln(\alpha) = 18.03 * \frac{10^3}{(T[°C] + 273.15)} - 32.42$$
$$with \ \alpha = \frac{\left(\frac{\delta^{18}O_c[‰VPDB]}{1000} + 1\right)}{\left(\frac{\delta^{18}O_w[‰VPDB]}{1000} + 1\right)} \ (\mathbf{2})$$
$$\delta^{18}O_w[‰VPDB] = 0.97002 * \delta^{18}O_w[‰VSMOW] - 29.98 \ (\mathbf{3})$$
To apply these formulae, it is assumed that carbonate is precipitated in equilibrium with the precipitation
fluid. Which carbonates are precipitated in equilibrium has long been subject to debate, and the
development of new techniques for measuring the carbonate-water system (e.g. clumped and dual-
clumped isotope analyses; Daëron et al., 2019; Bajnai et al., 2020) has led some authors to challenge
the assumption that equilibrium fractionation is the norm (see **Supplementary Discussion**). The
modular character of ShellChron allows the empirical transfer function to be adapted to the $\delta^{18}O_c$ record
or to the user's preference for alternative transfer functions by a small modification of the *d18O_model*
function. Future versions of the model will include more options for changing the transfer function (see
**Model description**).
As the name suggests, the ShellChron model was initially developed for application on $\delta^{18}O_c$ records
from marine calcifiers (e.g. mollusk shells and corals). ShellChron approximates the evolution of the
calcification temperature at which the carbonate is precipitated by a sinusoidal function (see **equation**
**4**, **Table 1** and **SI4**; *temperature_curve* function; visualized in **Fig. 4A** and **Fig S1**), a good approximation
of seasonal temperature fluctuations in most marine and terrestrial environments (Wilkinson and Ivany,
2002; Ivany and Judd, 2022). Variability in $\delta^{18}O_w$ is also comparatively limited in most marine
environments (except for regions with sea ice formation), making the model easy to use in these settings
(LeGrande and Schmidt, 2006; Rohling, 2013). Nevertheless, ShellChron includes the option to provide
*a priori* knowledge about $\delta^{18}O_w$, ranging from annual average values to detailed seasonal variability,
enabling the model to work in environments with more complex interaction between $\delta^{18}O_w$ and
temperature on the $\delta^{18}O_c$ record (see **equations 1 and 2**). This $\delta^{18}O_w$ data can be provided either as a
vector (with the same length as the data) or a single value (assuming constant $\delta^{18}O_w$) through the *d18Ow*
parameter in the *run_model* function.

$$T[°C] = T_{av} + \frac{T_{amp}}{2} \sin\left(\frac{2\pi * \left(t[d] - T_{pha} + \frac{T_{per}}{4}\right)}{T_{per}}\right) \quad (\mathbf{4})$$


# Table 1: Overview of model parameters

| Name | Description | Unit | Range |
|---|---|---|---|
| $T_{av}$ | Average temperature | °C | Variable, generally between 0°C–30°C |
| $T_{amp}$ | Temperature range (2*amplitude) | °C | Variable, generally <20°C |
| $T_{pha}$ | Phase of temperature sinusoid | d | 0–365 days |
| $T_{per}$ | Period of temperature sinusoid | d | 365 days by default |
| $G_{av}$ | Average growth rate | µm/d | Variable, generally between 0–100 µm/day |
| $G_{amp}$ | Range of growth rates | µm/d | Variable, generally <200 µm/day |
| $G_{pha}$ | Phase of growth rate sinusoid | d | 0–365 days |
| $G_{per}$ | Period of growth rate sinusoid | d | 365 days by default |
| $G_{skw}$ | Skewness factor of GR sinusoid | - | 0–100, with 50 meaning no skew |
| $D$ | Distance along the record | µm | Depends on archive |
| $t$ | Age | d | Depends on archive |
| $L_{win}$ | Length of sampling window | # | Depends on sampling resolution |
| $w$ | Weighting factor on sample | - | 0–1 |
| $i$ | Position relative to model window | - | $0–L_i$ |
| $I$ | Intercept of sinusoid ($T_{av}$ or $G_{av}$) | °C or µm/d | |
| $A$ | Amplitude of sinusoid $\left(\frac{T_{amp}}{2} \ or \ \frac{G_{amp}}{2}\right)$ | °C or µm/d | |
| $P$ | Period of sinusoid ($T_{per}$ or $G_{per}$) | d | |
| $\varphi$ | Phase of sinusoid ($T_{pha}$ or $G_{pha}$) | d | |


If marine $\delta^{18}O_c$ records represent one extreme on the spectrum of temperature versus $\delta^{18}O_w$ influence
on the $\delta^{18}O_c$ record, cave environments, in which $\delta^{18}O_c$ variability is predominantly driven by $\delta^{18}O_w$
variability in the precipitation fluid, represent the other extreme (Van Rampelbergh et al., 2014). In its
current form, ShellChron takes $\delta^{18}O_w$ as a user-supplied parameter to model temperature and growth
rate variability, but future versions will allow temperature to be fixed, while $\delta^{18}O_w$ becomes the modelled
variable. ShellChron's modular character makes it possible to implement this update without changing
the structure of the model. Application of ShellChron on $\delta^{18}O_c$ records from cave deposits will have to
be treated with caution, since drip water $\delta^{18}O_w$ seasonality (if present) cannot always be approximated
by a sinusoidal function and equilibrium fractionation in cave deposits is less common than in bio-
archives (Baldini et al., 2008; Daëron et al., 2011; Van Rampelbergh et al., 2014).
Besides temperature (or $\delta^{18}O_w$) seasonality, ShellChron models the growth rate of the archive to
approximate the $\delta^{18}O_c$ record (see **equation 5**, **Table 1** and **SI4**; *growth_rate_curve* function; visualized
in **Fig. 4B** and **Fig S2**). Since the growth rate in many carbonate archives varies seasonally, a quasi-
sinusoidal model for growth rate seems plausible (e.g. Le Tissier et al., 1994; Baldini et al., 2008; Judd
et al., 2018). However, as discussed in Judd et al. (2018), the occurrence of growth cessations (growth
rate = 0) and skewness in seasonal growth patterns calls for a more complex growth rate model that
can take these properties into account. Therefore, ShellChron uses a slightly modified version of the
skewed sinusoidal growth function described by Judd et al. (2018; **equation 5**). Note that the added
complexity of this function does not preclude the modelling of growth rate functions described by a
simple sinusoid (no skewness; $G_{skw}$ = 50) or even constant growth through the year ($G_{amp}$ = 0; see **Table**
**1**).
$$G[mm/yr] = G_{av} + \frac{G_{amp}}{2} \sin\left(\frac{2\pi * (t[d] - G_{pha} + G_{per} * S)}{P}\right)$$

$$with\ S = \begin{cases} \dfrac{100 - G_{skw}}{50}, & if\ t[d] - G_{pha} < G_{per}\dfrac{100 - G_{skw}}{100} \\ \dfrac{G_{skw}}{50}, & if\ t[d] - G_{pha} \geq G_{per}\dfrac{100 - G_{skw}}{100} \end{cases} (5)$$

Contrary to previous $\delta^{18}O_c$ growth models, ShellChron allows uncertainties on the input variables
(sampling distance and $\delta^{18}O_c$ measurements) as well as uncertainties of the full modelling approach to
be propagated, providing confidence envelopes around the chronology. Uncertainty propagation is
optional and can be skipped without compromising model accuracy. Standard deviations of uncertainties
on input variables (sampling distance and $\delta^{18}O_c$) can be provided by the user, while model uncertainties
are calculated from the variability in model results of the same datapoint obtained from overlapping
simulation windows (see *growth_model* function). Measurement errors are combined by projecting
Monte Carlo simulated values for sampling distance and $\delta^{18}O_c$ measurements on the modelled $\delta^{18}O_c$
curve through an orthogonal projection (**equation 6**; *mc_err_orth* function; visualized in **Fig S3**). The
measurement uncertainty projected on the distance domain is then combined with the model uncertainty
to obtain pooled uncertainties in the distance domain, which are propagated through the modelled $\delta^{18}O_c$
record to obtain uncertainties on the model result in the age domain. As a result of the sliding window
approach in ShellChron, model results for datapoints situated at the edges of windows are more
sensitive to small changes in the modelled parameters and therefore possess a larger model
uncertainty. To prevent these least certain model estimates from affecting the stability of the model,
model results are given more weight the closer they are situated towards the center of the model window
(see **equation 7** in *export_results* function; see also **Fig. S4**). This weighting is also incorporated in
uncertainty propagation through a weighted standard deviation (see **equation 8** from the *sd_wt*
function). Note that, despite the weighting solution, the size of uncertainties on the first and last positions
in the $\delta^{18}O_c$ record remains uncertain since they are based on a smaller number of overlapping windows
(see e.g. **Figure 3**).
$$\sigma_{meas} = \sqrt{\left(\frac{D_{sim} - \overline{D_{sim}}}{\sigma_D}\right)^2 + \left(\frac{\delta^{18}O_{sim} - \overline{\delta^{18}O_{sim}}}{\sigma_{\delta^{18}O}}\right)^2} \quad (6)$$

$$w[i] = 1 - \left|\frac{2i}{L_{window}} - 1\right| \quad (7)$$

$$\sigma_{weighted,i} = \sqrt{\frac{w_i * (x_i - \overline{w})^2}{\sum w[i] * \frac{N-1}{N}}} \quad (8)$$


## 3. Model description

ShellChron is organized as a series of functions that describe the step-by-step modelling process. A
schematic overview of the model is given in **Fig. 1**. A short **Test Case** is used to illustrate the modelling
steps in ShellChron. **Fig. 2** shows how the virtual **Test Case** was created from randomly generated
seasonal growth rate, $\delta^{18}O_w$ and temperature curves using the *seasonalclumped* R package (de Winter
et al., 2021a; see **Fig. 2**, **Supplementary Methods** and **SI2**) A wrapper function (*wrap_function*) is
included, which carries out all steps of the model procedure in succession to promote ease of use.

## Schematic overview of ShellChron model

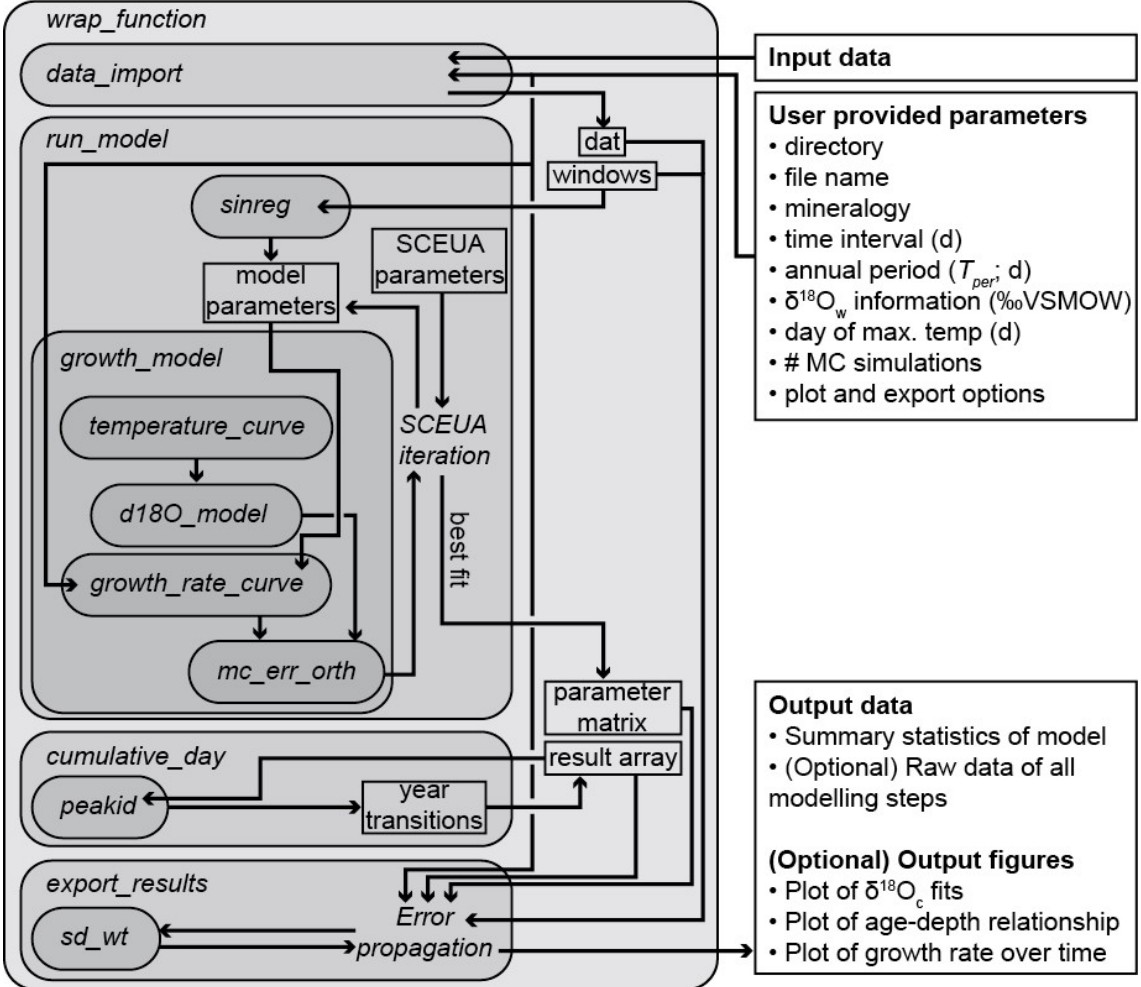


**Figure 1**: Schematic overview of ShellChron. Names in *italics* refer to functions (encapsulated in rounded rectangular boxes) and operations within functions. Rectangular boxes represent data. Arrows represent the flow of information between model components. Note that some operations are encapsulated in functions (e.g. *Error propagation* in *export results*) and that some functions are only used within other functions (e.g. *peakid* in *cumulative_day*). All data structures outside *wrap_function* represent input and output of the model. Detailed documentation of all functions and operations in ShellChron is provided in **SI1** (see also **Code availability**).

232

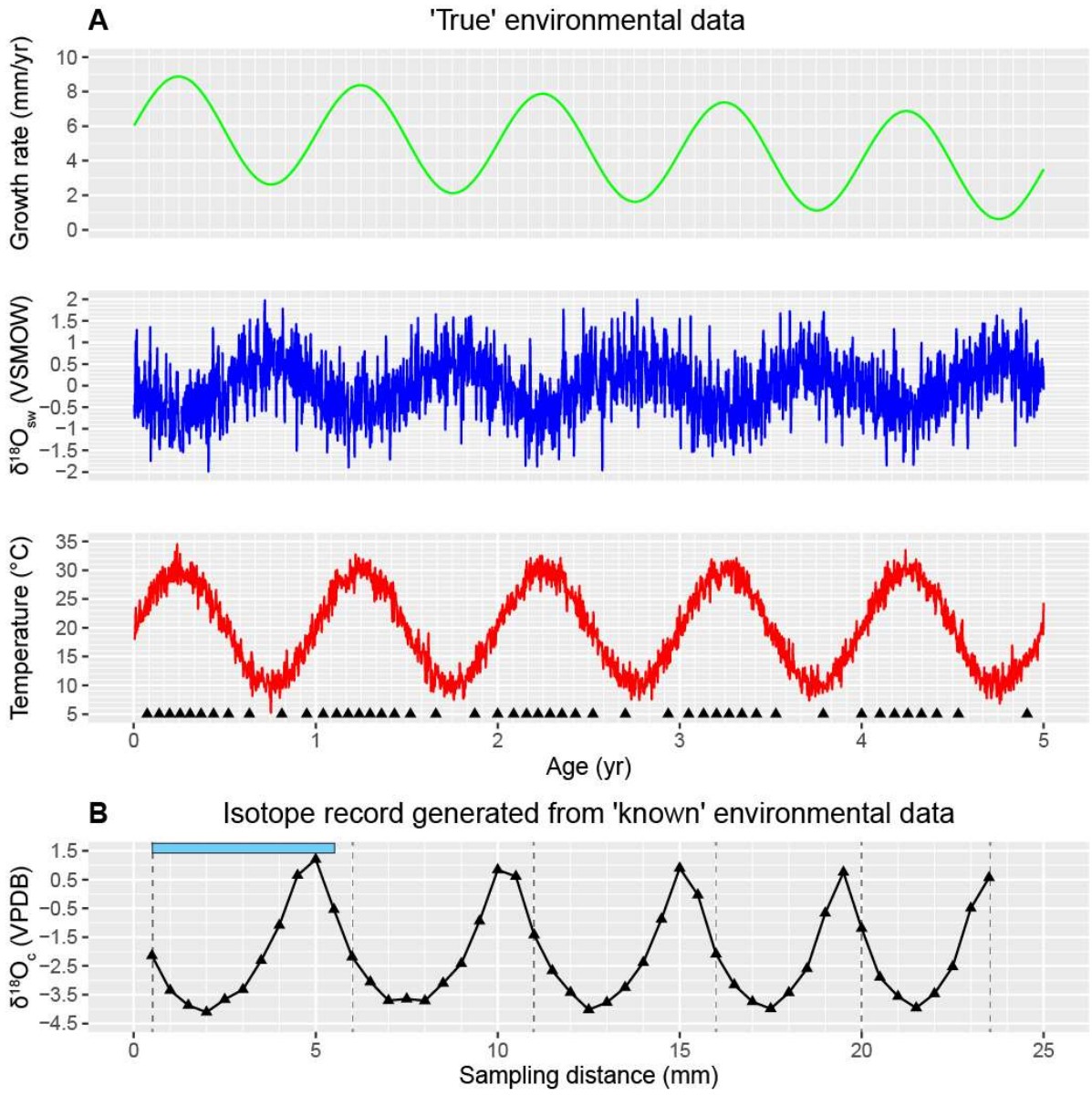

233

**Figure 2**: **A**) Plots of the growth rate (light green), $\delta^{18}O_w$ (blue) and temperature (red) records (in time domain) from which the **Test case** was produced. Black triangles on the bottom of the temperature plot indicate the ages of the samples taken from the record. **B**) The $\delta^{18}O_c$ record for the **Test Case** generated after equidistant sampling using the *seasonalclumped* package (de Winter et al., 2021a) with a sampling interval of 0.5 mm. Error bars on sampling distance (0.1 mm) and $\delta^{18}O_c$ (0.1‰) fall within the symbols. Vertical grey dashed lines indicate user-provided year markers and the blue bar on top of this plot shows an example of the width of a modelling window. See **Supplementary Methods** for details on producing the **Test case** $\delta^{18}O_c$ record and **SI3** for the R script used to generate the data.

Data is imported through the *data_import* function, which takes a comma-separated text file (CSV) with
the input data. Data files need to contain columns containing sampling distance ($D$, in μm) and $\delta^{18}O_c$
data (in ‰VPDB), a column marking years in the record (*yearmarkers*) and two optional columns
containing uncertainties on sampling distance ($\sigma(D)$, one standard deviation, in μm) and $\delta^{18}O_c$ ($\sigma(\delta^{18}O_c)$,
one standard deviation, in ‰) respectively (see example in **SI2** and **Figure 3**). The function uses the
year markers (third column) as guidelines for defining the minimum length of the model windows to
ensure that all windows contain at least one year of growth. By default, consecutive windows are shifted
by one datapoint, yielding a total number of windows equal to the sample size minus the length of the
last window. While year markers are required for ShellChron to run (otherwise no windows can be
defined), the result of the model does not otherwise depend on user-provided year markers, instead
basing the age result purely on simulations of the $\delta^{18}O_c$ data.

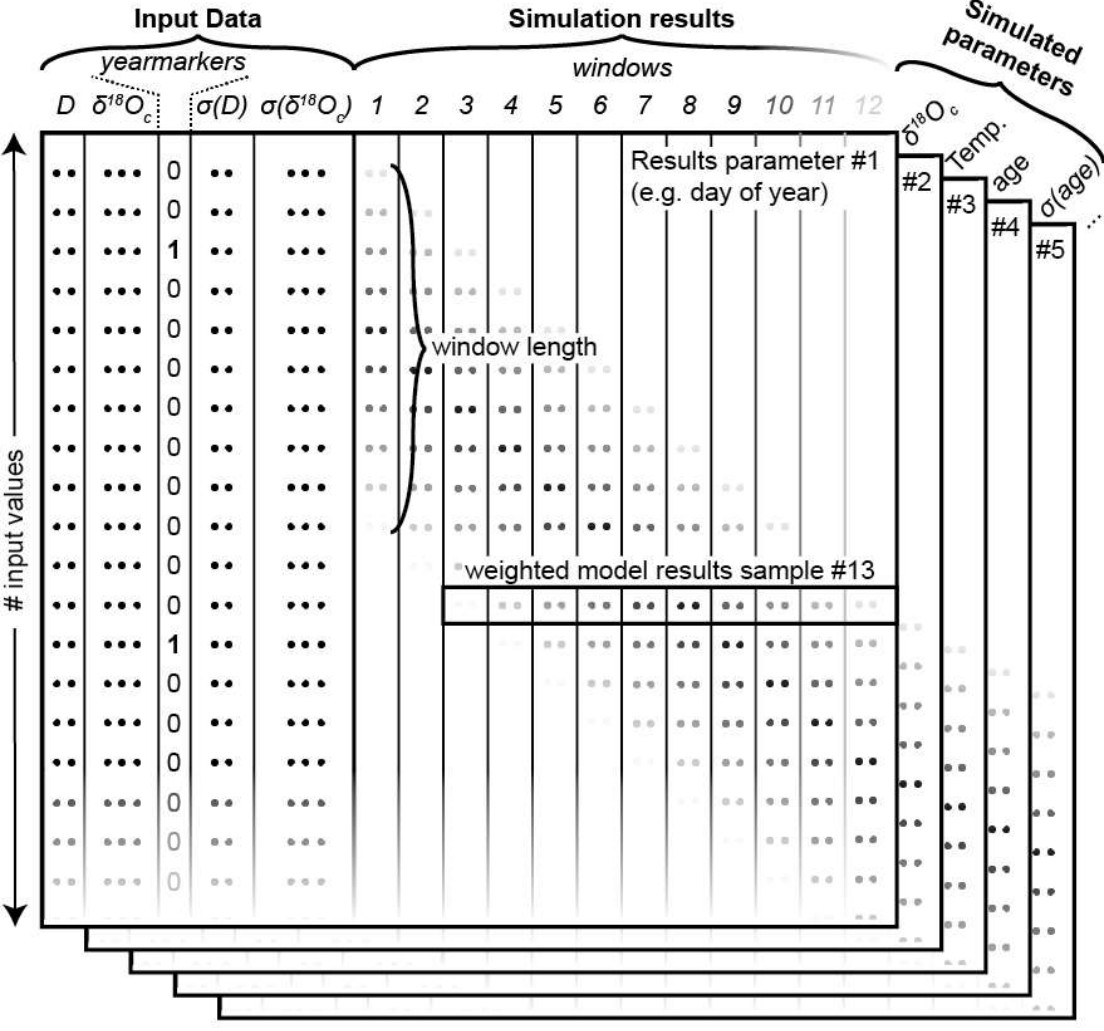

**Figure 3:** Schematic overview of the structure of the result array in which ShellChron stores the raw

results of each model window. Data is stored in three dimensions: The sample number (rows in the

figure), the window number (columns in the figure) and the number of modelled parameters

(represented by the stacked table "sheets" in the figure). Note that the first 5 columns of each "sheet"

represent the user-provided input data (see example in **SI2**), and that the model result data starts from

column 6. The window length is determined by the user-provided indication of year transitions (column

3). Rows of dots in the figure are placeholders for (input or result) values. Shading of these dots in the

window columns indicate differential weighting of modelled values in function of their location relative

to the sliding window. The horizontal box shows how these weighting factors within each sample

window (in vertical direction) result in weighting of different estimates of modelled parameters for the

same data point (in horizontal direction). Shading of input data and window number towards the

bottom and right edge of the figure, respectively, indicates that the number of input values (and thus

simulation windows) is only limited to the length of the input table and may therefore continue
indefinitely (at the expense of longer computation times, see **Fig. 8** in **Model performance**).
The core of the model consists of simulations of overlapping subsamples (windows) of the sampling
distance and $\delta^{18}O_c$ data described by the *run_model* function (see **Fig. 1 and 3**). Data and window sizes
are passed from *data_import* onto *run_model* along with user-provided parameters (e.g. $\delta^{18}O_w$
information; see **Fig. 1**). *run_model* loops through the data windows and calls the *growth_model*
function, which fits a modelled $\delta^{18}O_c$ vs. distance curve through the data using the SCEUA optimization
algorithm (see Duan et al., 1992; see example in **Fig 4**). The simulated $\delta^{18}O_c$ curve is produced through
a combination of a temperature sinusoid (*temperature_curve* function; see **equation 4**, **Fig. 4A** and **Fig.**
**S1**) and a skewed growth rate sinusoid (*growth_rate_curve*; see **equation 5**, **Fig. 4B** and **Fig. S2**), with
temperature data converted to $\delta^{18}O_c$ data through the *d18O_model* function (**equation 1 and 2**; **Fig.**
**4A**).

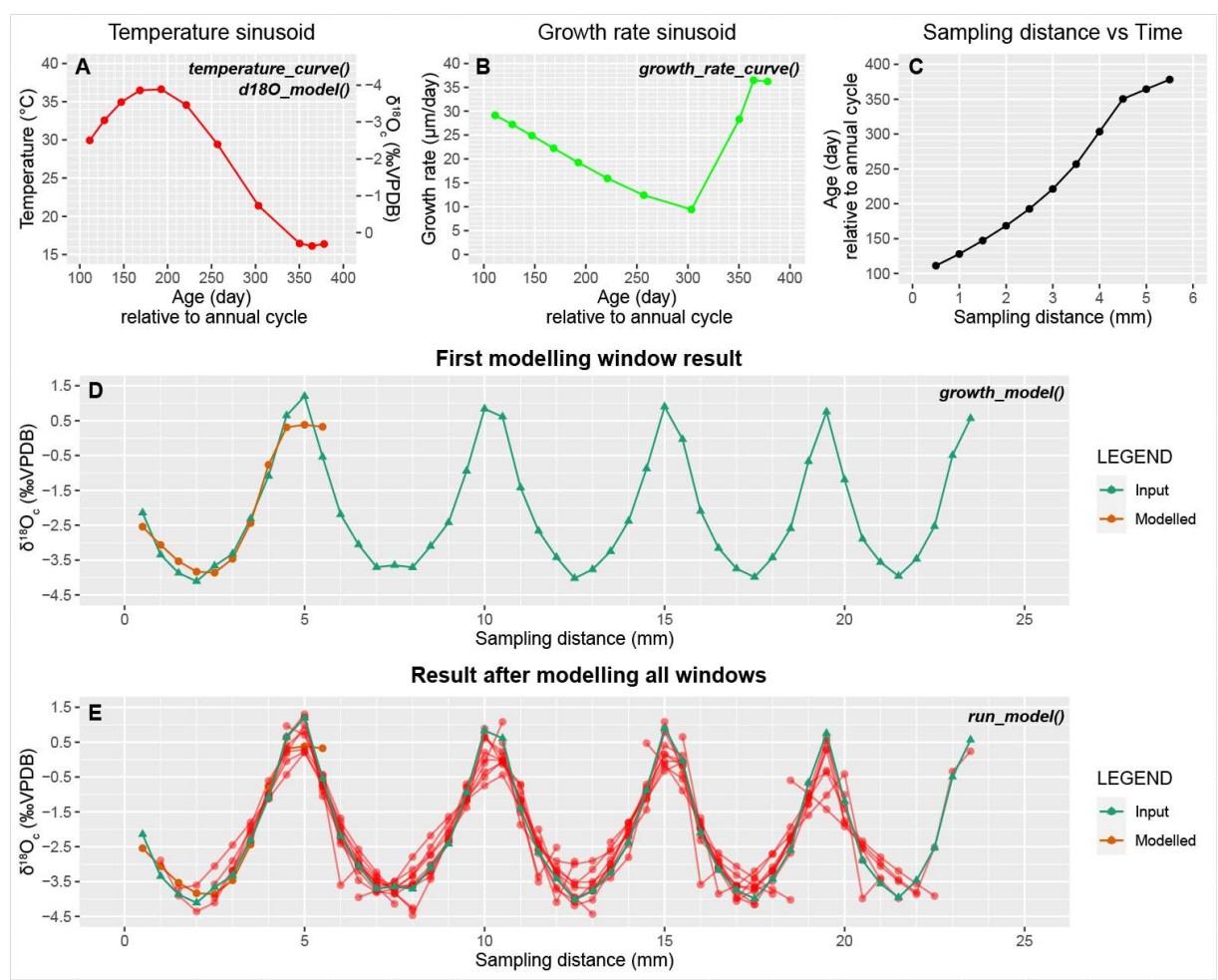


**Figure 4**: Showing the steps taken to simulate $\delta^{18}O_c$ data in the *run_model* function on the **Test case**.
**A**) Temperature sinusoid used to approximate $\delta^{18}O_c$ data in the first modelling window (see **D**), produced using a combination of *temperature_curve* and *d18O_model* functions. Symbols indicate the positions of $\delta^{18}O_c$ samples on the temperature curve, with estimated $\delta^{18}O_c$ values shown on the secondary axis (right). **B**) Skewed growth rate sinusoid fit to the $\delta^{18}O_c$ data using the *growth_rate_curve* function. Note the shift towards steeper growth rate increase around the 300th model day (autumn season in this example). See **Fig. S2** for a detailed description of the growth rate sinusoid. **C**) The modelled age-distance relationship for this window after fitting $\delta^{18}O_c$ data, resulting from aligning the estimated age of samples (x-axes on **A**) with the distance in sampling direction (x-axis in **D**) using the cumulative growth rate function (**B**). **D**) $\delta^{18}O_c$ profile of the **Test case** (green) with the $\delta^{18}O_c$ curve of the first modelling window (red), which results from the combination of temperature (**A**) and growth rate (**B**) sinusoids, plotted on top (*growth_model* function). **E**) Result after simulating the full $\delta^{18}O_c$ profile of the **Test case** (green) using *run_model*, with the $\delta^{18}O_c$ curves of individual modelling windows shown in red.

By default, starting values for the parameters describing temperature and growth rate curves are
obtained by estimating the annual period ($P$) through a spectral density estimation and applying a
linearized sinusoidal regression through the $\delta^{18}O_c$ data (*sinreg* function; see **equation 9**). It is possible
to skip this sinusoidal modelling step through the "*sinfit*" parameter in the *run_model* function, in which
case the starting value for the annual period is set equal to the width of the model window. In addition,
*growth_model* takes a series of parameters describing the method for SCEUA optimization (see Duan
et al., 1992; Judd et al., 2018) and the upper and lower bounds for parameters describing temperature
and growth rate curves (see **SI4**). Parameters for the SCEUA algorithm (*iniflg*, *ngs*, *maxn*, *kstop*, *pcento*
and *peps*) in the *run_model* function may be modified by the user to reach more desirable optimization
outcomes. The effect of changing the SCEUA parameters on the model result for the **Test case** is
illustrated in **section 4.1** (see **Fig. 5**). If uncertainties on sampling distance and $\delta^{18}O_c$ data are provided,
*growth_model* calls the *mc_err_orth* function to propagate these errors through the model result (see
**equation 6** and **Fig S3**).

$$\delta^{18}O_c[‰VPDB] = I + \frac{A}{2}\sin\left(\frac{2\pi * \left(D - \varphi + \frac{P}{4}\right)}{P}\right),$$

$$linearized\ as: \delta^{18}O_c[‰VPDB] = a + b\sin\left(\frac{2\pi}{P} * D\right) + c\cos\left(\frac{2\pi}{P} * D\right),$$

$$with\ I = a; A = \sqrt{b^2 + c^2}\ and\ \varphi = P * \left(0.25 - \frac{\cos^{-1}\left(\frac{b}{A}\right)}{2\pi}\right)\ \mathbf{(9)}$$

The *run_model* function returns an array listing day of the year (1–365), temperature, $\delta^{18}O_c$, growth rate
and (optionally) their uncertainty standard deviations as propagated from uncertainties on the input data
("result array"; see **Fig. 3** and **SI5**). Note that the default length of the year (*Tper* and *Gper*) is set at 365
days, but that these parameters can be modified by the user in *run_model*. In addition, a matrix
containing the optimized parameters of temperature and growth rate curves is provided, yielding
information about the evolution of mean values, phases, amplitudes, and skewness of seasonality in
temperature and growth rate along the record ("parameter matrix", see **Fig. 1** and **SI6**). To construct an
age model for the entire record, the modelled timing of growth data, expressed as day relative to the
365-day year, is converted into a cumulative time series listing the number of days relative to the start
of the first year represented in the record (rather than relative to the start of the year in which the
datapoint is found). This requires year transitions (transitions from day 365 to day 1) to be recognized
in all the model results. The *cumulative_day* function achieves this by aggregating information about
places where the beginning and end of the year is recorded in individual window simulations and
applying a peak identification algorithm (*peakid* function) to find places in the record where year
transitions occur (see **Supplementary Methods**). Results of the timing of growth for each sample (in
day of the year) are converted to a cumulative time scale using their positions relative to these
recognized year transitions (**Supplementary Methods**).
In a final step (described by the *export_results* function), the results from overlapping individual
modelling windows are combined to obtain mean values and 95% confidence envelopes of the result
variables (age, $\delta^{18}O_c$, $\delta^{18}O_c$-based temperatures and growth rates) for each sample in the input data. If
uncertainties on the input variables were provided, these are combined with uncertainties on the
modelling result calculated from results of the same datapoint on overlapping data windows by pooling
the variance of the uncertainties (**equation 10**). Throughout this merging of data from overlapping
windows, results from datapoints on the edge of windows are given less weight than those from
datapoints near the center of a window (see **equation 7** and **Fig. S4**). This weighting procedure corrects
for the fact that datapoints near the edge of a window are more susceptible to small changes in the
model parameters and are therefore less reliable than results in the center of the window. Finally,
summaries of the simulation results and the model parameters including their confidence intervals are
exported as comma-separated (CSV) files. In addition, *export_results* supports optional exports of
figures displaying the model results and files containing raw data of all individual model windows
(equivalent to "sheets" of the result array, see **Fig. 3** and **SI5**).

$$VAR_{pooled} = \frac{\sum_i((N_i-1)*VAR_i*w_i)}{\sum_i(N_i)-n} \textbf{(10)}$$

in which *w* = weight of the individual reconstructions, *N* is the sample size and *n* is the number of
reconstructions (indexed by *i*) that is combined

## 4. Model performance

The performance of ShellChron was first tested on three virtual datasets:

1. The short **Test case** used to illustrate the model steps above (see **Fig. 2** and **4**; **SI7**)

2. A $\delta^{18}O_c$ record constructed from a simulated temperature sinusoid with added stochastic noise (**Case 1**; **SI8**)

3. A record based on a known high-resolution sea surface temperature and salinity record measured on the coast of Texel island in the tidal basin of the Wadden Sea (North Netherlands; **Texel**, see details in **SI9** and de Winter et al., 2021a and **Supplementary Methods**).

Firstly, the effect of varying parameters in the SCEUA algorithm is tested on the **Test Case** (**Fig. 5**). Then, full model runs on **Case 1** and **Texel** are evaluated in terms of model performance (**Fig. 6**). In addition to the three test cases, three modern carbonate $\delta^{18}O_c$ records were internally dated using ShellChron (see **Fig. 7**): a tropical stony coral (*Porites lutea*; hereafter: **coral**) from the Pandora Reef (Great barrier Reef, NE Australia; Gagan et al., 1993; see **SI10**), a Pacific oyster shell (*Crassostrea gigas*; hereafter: **oyster**) from List Basin in Denmark (Ullmann et al., 2010; see **SI10**) and a temperate zone speleothem from Han-sur-Lesse cave (Belgium; hereafter: **speleothem**; see Vansteenberge et al., 2019; see **SI10**). Finally, ShellChron's performance in terms of computation time and accuracy is compared to that of the most comprehensive pre-existing $\delta^{18}O_c$-based age model (GRATAISS model by Judd et al., 2018) on simulated temperature sinusoids of various length and sampling resolutions to which stochastic noise was added (*sensu* **Case 1**; de Winter et al., 2021a; see **Fig. 8** and **SI11**). The latter also demonstrates the scalability of ShellChron and its application on a variety of datasets. Timing comparisons were carried out using a modern laptop (Dell XPS13–7390; Dell Inc., Round Rock, Tx, USA) with an Intel Core i7 processor (8 MB cache, 4.1 GHz clock speed, 4 cores, Intel Corporation, Santa Clara, CA, USA), 16 GB LPDDR3 RAM and an SSD drive running Windows 10. Note that ShellChron was built and tested successfully on Mac OS, Fedora Linux and Ubuntu Linux as well.

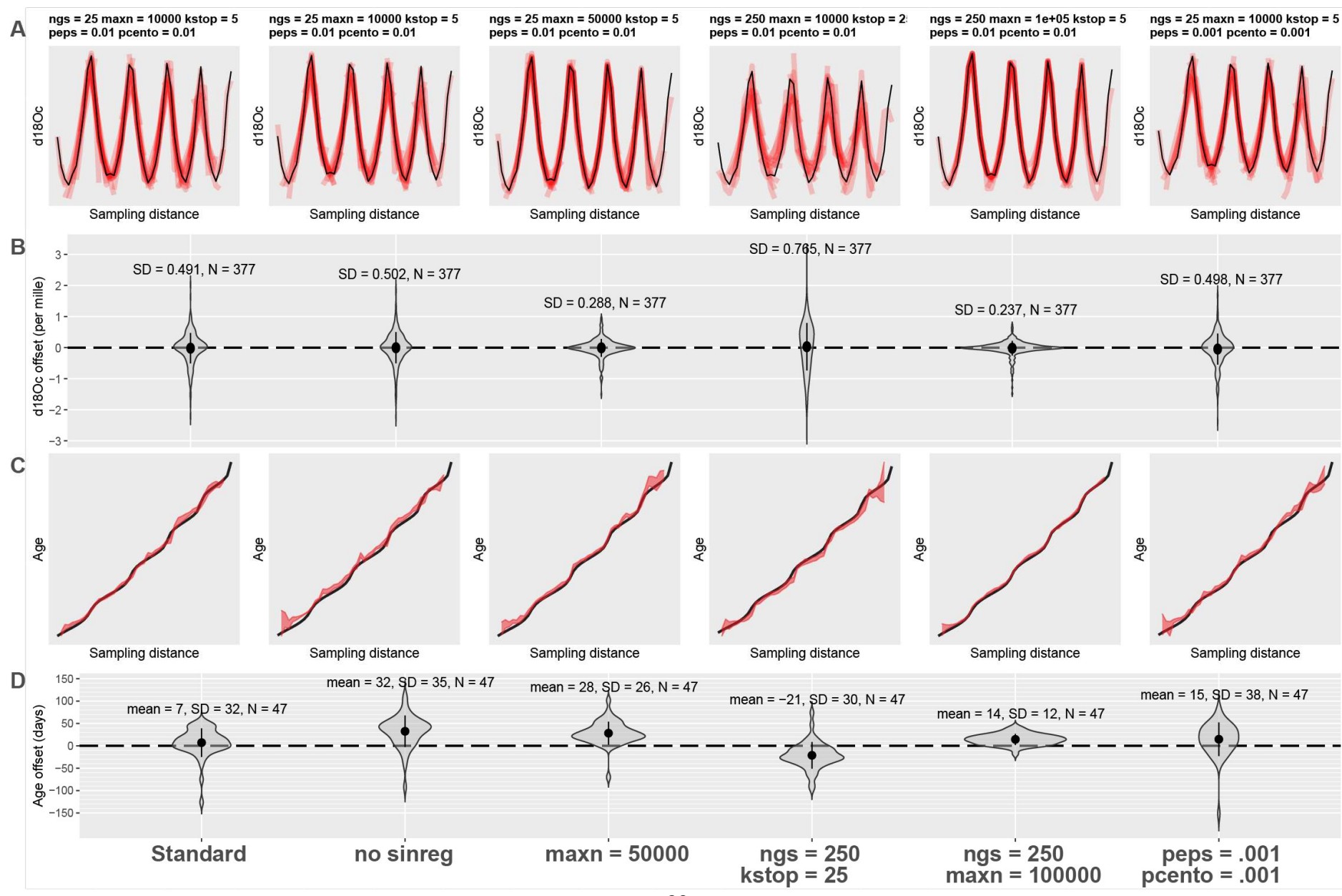

**Figure 5:** Result of testing ShellChron with various combinations of SCEUA parameters and sinusoidal regression on the **Test case** dataset (see **Fig. 2**). The leftmost plots illustrate performance of ShellChron under default SCEUA parameters. Plots to the right show various combinations of parameters that deviate from the default (see labels on top and bottom of plot) **A)** Fits of the model $\delta^{18}O_c$ curves (red) with the data (black). **B)** Violin plots showing the distribution of modelled $\delta^{18}O_c$ offset from the data. **C)** Age-distance plots showing modelled (red) and known (black) age-depth relationships for each scenario. **D)** Violin plots showing the distribution of age offsets from the known age-depth relationship. SD = standard deviation, N = number of datapoints, sinres = sinusoidal regression, maxn, ngs, kstop, peps and pcento are SCEUA parameters (see Duan et al., 1992 and explanation in **section 4.1**). Data on test results is provided in **SI11**.

**4.1 Testing model parameters**

Testing different combinations of modelling parameters (**Fig. 5**) shows that, while the results of ShellChron can improve beyond the default SCEUA parameters and sinusoidal regression, care must be taken to evaluate the effect of changing modelling parameters on both the $\delta^{18}O_c$ fit and the age-distance relationship. Comparative testing on the **Test case** (**Fig. 5**) shows that sinusoidal regression has a negligible influence on the success of ShellChron fitting the $\delta^{18}O_c$ curve (**Fig. 5A-B**; standard deviation on $\delta^{18}O_c$ is 0.49‰ with sinusoidal regression and 0.50‰ without). However, ShellChron with sinusoidal regression performs better in terms of age approximation, with a mean age offset of only 7 ± 32 days with sinusoidal regression against 32 ± 35 days without (**Fig. 5C-D**). Age-distance plots (**Fig. 5C**) show that the model without sinusoidal fit shows a phase offset with respect to the known age-distance relationship, resulting in overestimation of the age for much of the record. Sinusoidal regression probably results in better initial parameter estimation, which helps to avoid phase offsets like the one shown in **Fig. 5**. For the remainder of the tests, sinusoidal regression was enabled.

The remainder of the tests show that the main bottleneck towards better $\delta^{18}O_c$ fit optimization is the maximum number of function evaluations allowed within a single modelling cycle (maxn; see **Fig. 5**). Increasing the other SCEUA parameters, such as the number of complexes in the SCEUA routine (ngs), the number of shuffling loops that should show a significant change before convergence (kstop) and the thresholds for significant change in parameter value (peps) or result value (pcento) does not improve the result if the SCEUA algorithm is not allowed more processing time (maxn). In fact, **Fig. 5**

shows that increasing these SCEUA parameters can actually result in a deterioration of the $\delta^{18}O_c$ fit
and higher uncertainty on the age result (**Fig. 5B and D**). A fivefold increase in maxn (maxn = 50000)
almost halves the standard deviation on $\delta^{18}O_c$ residuals (from 0.49‰ to 0.29‰; **Fig. 5B**) and
decreases the standard deviation on the age model offset from 32 to 26 days (**Fig. 5D**). A combination
of a tenfold increase in function evaluations with an equal multiplication of the number of complexes in
the SCEUA routine (ngs; see details in Duan et al., 1992) results in a further reduction of standard
deviations on $\delta^{18}O_c$ (0.23‰) and age result (12 days). These tests show that returns in terms of model
precision quickly diminish with increasing processing time. Since the total modelling time linearly
scales with the number of function evaluations, this tradeoff towards lower standard deviation on the
modelling result is costly. These function evaluations are repeated in each modelling window, so the
cost in terms of extra processing time can increase quickly, especially for larger $\delta^{18}O_c$ datasets. In
addition, in this situation the mean model offset (accuracy of the model; 7 days, 28 days and 14 days
for maxn of $1.0 * 10^4$, $5.0 * 10^4$ and $1.0 * 10^5$ respectively; **Fig. 5D**) does not significantly improve with
increasing number of function evaluations. Based on these results, the default maxn parameter in
ShellChron was set to $10^4$ to compromise between keeping modelling times short while retaining high
model accuracy. However, specific datasets may benefit from an increase in modeling time, so case-
by-case assessment of the optimal SCEUA parameters is recommended. A detailed evaluation of the
total modelling time in a typical $\delta^{18}O_c$ dataset is discussed in **section 4.4**.

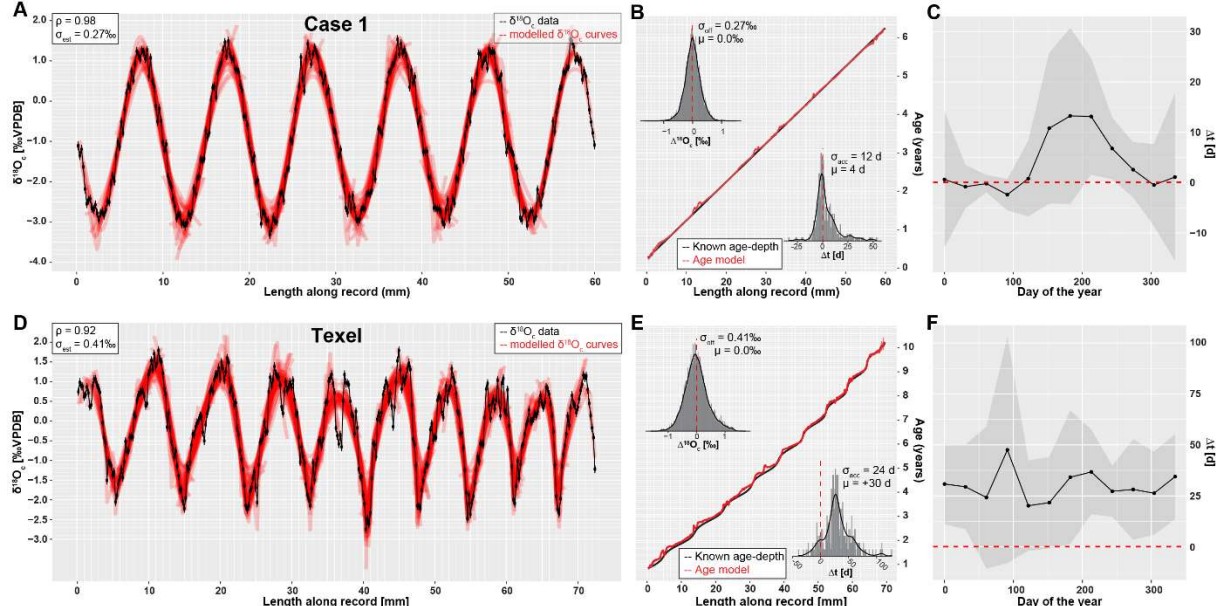

**Figure 6**: Result of applying ShellChron on two virtual datasets: **Case 1** (top, see **SI8**) and **Texel**, (bottom, see **SI9**). Leftmost panels (**A** and **D**) show the model fit of individual sample windows (red) on the data (black, including horizontal and vertical error bars), with in the top left Spearman's correlation coefficients (ρ) and standard deviations on the $\delta^{18}O_c$ estimate ($\sigma_{est}$). Middle panels (**B** and **E**) show the resulting age model (red, including shaded 95% confidence level) compared with the known age-distance relationship of both records. Histograms in the top left of age-distance plots show the offset between modelled and measured $\delta^{18}O_c$ (as visualized in panels **A** and **D**) with standard deviations of the $\delta^{18}O_c$ offset ($\sigma_{off}$) and offset averages (μ). Histograms in the bottom right of age-distance plots show the offset between modelled and known ages (in days) of each datapoint, including standard deviations on the age accuracy ($\sigma_{acc}$) and mean age offset (μ). Rightmost panels (**C** and **F**) highlight age offsets binned in 12 monthly time bins based on their position relative to the annual cycle to illustrate how accuracy varies over the seasons. Grey envelopes indicate 95% confidence levels on the monthly age offset within these monthly time bins. The horizontal red dashed line indicates no offset (modelled age is equal to the known age of the sample).

## 4.2 Artificial carbonate records

Results of running ShellChron on the **Test case** (**Fig. 4**), **Case 1** and **Texel** datasets (**Fig. 6**) show that modelled $\delta^{18}O_c$ records in individual windows closely match the data. On the level of individual windows, inter-annual growth rate variability is more difficult to model than the temperature sinusoid, especially when sampling resolution is limited and at the beginning and end of the record (**Fig. 4B**). However, after overlapping multiple windows, the accuracy of ShellChron improves significantly (**Fig. 4E**). Note that in **Fig. 4A-C**, the length of the first model window (difference in age between first and 11th datapoint) is less than 365 days, because the 12th datapoint, which occurs exactly 1 year after the first point, is not part of the window. A summary of ShellChron performance statistics is given in **Table 2**. In all virtual datasets, $\delta^{18}O_c$ estimates are equally distributed above and below the $\delta^{18}O_c$ data ($\overline{\Delta^{18}O_c} = 0.0$ ‰; Spearman's $\rho$ of 0.94, 0.98 and 0.92 for **Test case**, **Case 1** and **Texel** datasets respectively). Age offsets vary slightly over the seasons, but the difference between monthly time bins is not statistically significant on a 95% confidence level (**Fig. 6C** and **F**; see also **SI12**). The fact that seasonal bias in age offset is absent in the **Texel** dataset, which is skewed towards growth in the winter season and includes relatively strong seasonal variability in $\delta^{18}O_w$, shows that ShellChron is not sensitive to such subtle (though common) variability in growth rate or $\delta^{18}O_w$. In general, ShellChron's mean age assignment is accurate on a monthly scale (age offsets of 4 ± 12 d and +30 ± 24 d for **Case 1** and **Texel** datasets respectively). However, age results in individual months do sometimes show significant offsets from the known value (e.g. **Fig. 6C** and **6F**). This is most notable in **Case 1**, where accuracy of the age model decreases near the extreme values of the $\delta^{18}O_c$ curve (**Fig. 6B-C**). This occurs because in these places the model is most sensitive to stochastic noise (simulated uncertainty) on the $\delta^{18}O_c$ value. A small random change in the $\delta^{18}O_c$ value at the minima or maxima of the $\delta^{18}O_c$ curve thus results in a large change in the model fit of the $\delta^{18}O_c$ curve, resulting in a seasonally non-uniform decrease in the accuracy of the model, as is evident from the skewed $\Delta^{18}O_c$ distribution in **Figure 6B-C**. The sampling resolution in the **Texel** data decreases near the end of the record (see **SI9**), but this does not result in reduced age model accuracy. If anything, the age of **Texel** samples is better approximated near the end of the record, and age offsets are larger in the central part of the record (~30-50 mm; **Fig. 6E**). The lower accuracy in the third to fifth year of the **Texel** record is likely a result of the sub-annual variability in the record that is superimposed on the seasonal cycle. The lower sampling resolution later in the record mutes this variability and illustrates that higher sampling resolutions do not necessarily result in better age models.

The constant offset of the modelled age of the **Texel** sample from the known age is a result of the way
the model result was aligned to start at zero for comparison with the known age (**Fig. 6F**). This was
done by adding the offset from zero of the modelled age of the first datapoint in the record to the entire
record, thereby defining an arbitrary reference point which is sensitive to the uncertainty on the age of
the first sample (see also **Oyster** and **Speleothem** results in **Fig. 7B-C**). Note that this alignment issue
does not play a role in fossil data, where model results can be aligned to growth marks in the carbonate
(e.g. shell growth breaks or laminae) and that it does not affect the seasonal alignment of proxy binned
into monthly sample bins.

# Table 2: Overview of datasets and model results

| Dataset | Resolution | Length | $\delta^{18}O_c$ seasonal range | Complications |
|---|---|---|---|---|
| Test case | 7-12 yr$^{-1}$ | 5 yr | ~5‰ | Variable $\delta^{18}O_w$, Variable GR |
| Case 1 | 50 yr$^{-1}$ | 6 yr | ~4.3‰ | None |
| Texel | 26–45 yr$^{-1}$ | 10 yr | ~4‰ | Variable $\delta^{18}O_w$, Variable GR |
| Coral | 30–49 yr$^{-1}$ | 6 yr | ~1.7‰ | Variable GR |
| Oyster | 23–45 yr$^{-1}$ | 3.5 yr | ~3‰ | Variable $\delta^{18}O_w$, Variable GR |
| Speleothem | 4–13 yr$^{-1}$ | 14 yr | ~0.5‰ | Variable $\delta^{18}O_w$, Variable GR, Non-sinusoidal $\delta^{18}O_c$-forcing |

| Dataset | $\delta^{18}O_c$ offset ($\pm1\sigma$) | Age offset ($\pm1\sigma$) | Spearman's $\rho$ | Observations |
|---|---|---|---|---|
| Test case | 0.0 ± 0.49 ‰ | 7 ± 32 d | 0.94 | Slightly out of phase |
| Case 1 | 0.0 ± 0.27‰ | 4 ± 12 d | 0.98 | - |
| Texel | 0.0 ± 0.41‰ | 30 ± 24 d | 0.92 | - |
| Coral | 0.0 ± 0.14‰ | 12 ± 28 d | 0.97 | - |
| Oyster | 0.0 ± 0.39‰ | -15 ± 43 d | 0.91 | Reduced accuracy near growth stops |
| Speleothem | 0.0 ± 0.08‰ | -114 ± 59 d | 0.92 | Susceptible to phase offsets; Only reliable on inter-annual scale |


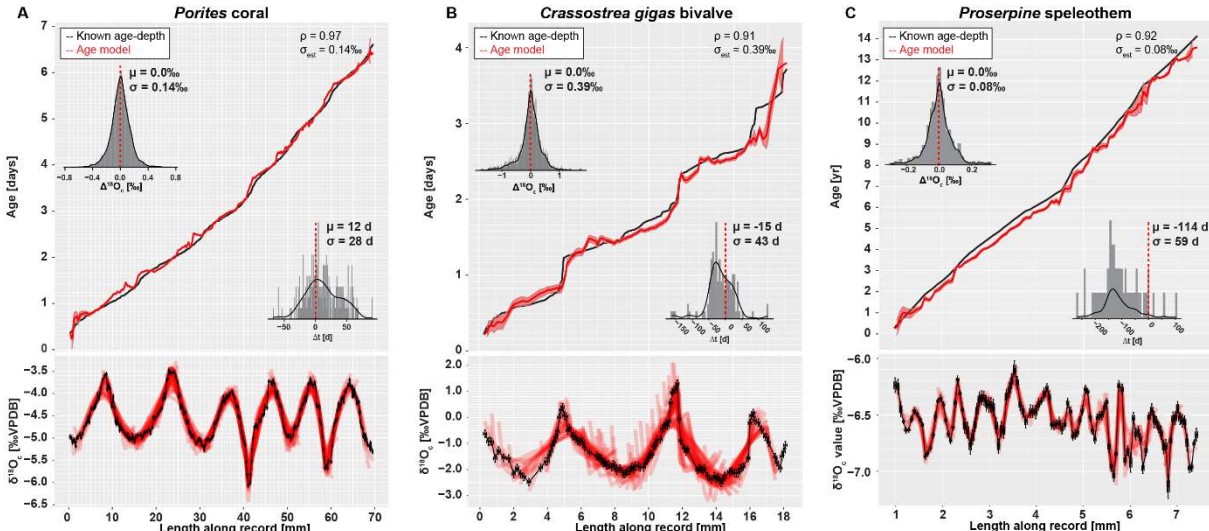

Figure 7: Overview of model results for the three test datasets from real carbonate archives: (**A**) **coral**, (**B**) **oyster** and (**C**) **speleothem**. Lower panels indicate the fit of individual model windows (in red) with the data (in black) while upper panels show the age model (in red) compared to the "true" age-distance relationship with histograms showing model accuracy (in days, top left) and model fit ($\delta^{18}O_c$ offset in ‰, bottom right). Color scheme follows **Figure 3**. Note that the true age-distance relationship is not known for these natural records, but is estimated using known growth seasonality (**coral**), comparison with *in situ* temperature and salinity measurements (**oyster)** or simply by interpolating between annual growth lines (**speleothem**). See **Supplementary Methods** for details and **SI10** for raw data.

**4.3 Natural carbonate records**

Results of modelling natural carbonate records (**Fig. 7**; **Table 2**; see also **SI10**) illustrate the effectiveness of ShellChron on various types of records. Performance clearly depends on the resolution of the record and the regularity of seasonal variability contained within. As in the virtual datasets, modelled $\delta^{18}O_c$ successfully mimic $\delta^{18}O_c$ data in all records ($\overline{\Delta^{18}O_c} = 0.0$; Spearman's $\rho$ of 0.97, 0.91 and 0.92 for **coral**, **oyster** and **speleothem** respectively). No consistent seasonal bias is observed in $\Delta^{18}O_c$ and model accuracy ($p > 0.05$; see **Table 2** and **SI12**), despite significant (seasonal and inter-annual) variability contained in the records (especially in **oyster** and **speleothem** records). When comparing the accuracy of these records, it must be noted that the "known" age of the samples in these natural carbonates is not known. Model results are instead compared with age models constructed using conventional techniques such as matching $\delta^{18}O_c$ profiles with local temperature and/or $\delta^{18}O_w$ variability (**oyster** and **coral** records) or even merely by linear interpolation between annual markers in the record (**speleothem** record; see **Supplementary Methods**). Despite this caveat, testing results clearly show that the least complicated record (**coral**; **Fig. 7A**), characterized by minimal variability in $\delta^{18}O_w$ and growth rate and a high sampling density, has the best overall model result ($\Delta^{18}O_c = 0.0 \pm 0.14$ compared to a ~1.7‰ seasonal range; $\rho = 0.97$; $\Delta t = 12 \pm 28$ d; see **Table 2**). The **oyster** record (**Fig. 7B**), which has strong seasonal variability in growth rate and $\delta^{18}O_{sw}$ also yields a reliable age model ($\Delta^{18}O_c = 0.0 \pm 0.39$ compared to a ~3‰ seasonal range; $\rho = 0.91$; $\Delta t = -15 \pm 43$ d; see **Table 2**). On closer inspection, the age within the **oyster** record is clearly more difficult to model than within the **coral**, due in part to the higher variability of $\delta^{18}O_c$ values superimposed on the seasonal cycle, the sharp growth cessations in the winters (high $\delta^{18}O_c$ values) and the variability in sampling resolution within the record. The latter causes the first growth year of the **oyster** record to be less accurately modelled (**Fig. 7B**) while the variability in $\delta^{18}O_c$ causes the edges of some modelling windows to predict steep increases or decreases in $\delta^{18}O_c$ (vertical "offshoots" in modelled $\delta^{18}O_c$; **Fig. 7B**). Note that the low weighting of the edges of modelling windows combined with the high overall sampling resolution in the **oyster** record minimizes the effect of these "offshoots" on the accuracy of the model. The **speleothem** record (**Fig. 7C**), plagued by lower sampling resolution, large inter-annual $\delta^{18}O_c$ variability, restricted $\delta^{18}O_c$ seasonality and a lack of clearly seasonal $\delta^{18}O_c$ forcing, yields the least reliable model result ($\Delta^{18}O_c = 0.0 \pm 0.08$‰ compared to a ~0.5‰ seasonal range; $\rho = 0.92$; $\Delta t = -114 \pm 59$ d; see **Table 2**). Note that the accuracy figure provided for the **speleothem** record is based on comparison with an age model relying on linear

interpolation between annual growth lines. This assumption of the age-distance relationship is almost
certainly erroneous, since drip water supply to (and therefore growth in) speleothems has been shown
to vary seasonally (e.g. Baldini et al., 2008), including at the very site the **speleothem** data derives from
(Han-sur-Lesse cave, Belgium; Van Rampelbergh et al., 2014; Vansteenberge et al., 2019). However,
since no reliable information is available on sub-annual variability in growth rates in this record,
ShellChron results cannot be validated at the sub-annual scale in this case. The high age offset (-114
days) in the **speleothem** model result is a consequence of the assumption in ShellChron that the highest
temperature (lowest $\delta^{18}O_c$ value) recorded in each growth year happens halfway through the year (day
183) and the alignment of the modelled age with the "known" age for this record (see discussion of **Texel**
results in 4.2). While the assumption about the phase of the temperature sinusoid is approximately valid
for temperature-controlled $\delta^{18}O_c$ records (see **Fig. 6** and **7**), it is problematic for speleothems, in which
$\delta^{18}O_c$ is often dominated by the $\delta^{18}O_w$ of drip water, which may not be lowest during the summer season
(see Van Rampelbergh et al., 2014). The timing of the $\delta^{18}O_c$ minimum can be set in the *run_model*
function using the *t_maxtemp* parameter. Note that changing *t_maxtemp* does not affect relative dating
within the $\delta^{18}O_c$ record, but, if set correctly, results in a phase shift of the age model result into better
alignment with the seasonal cycle.

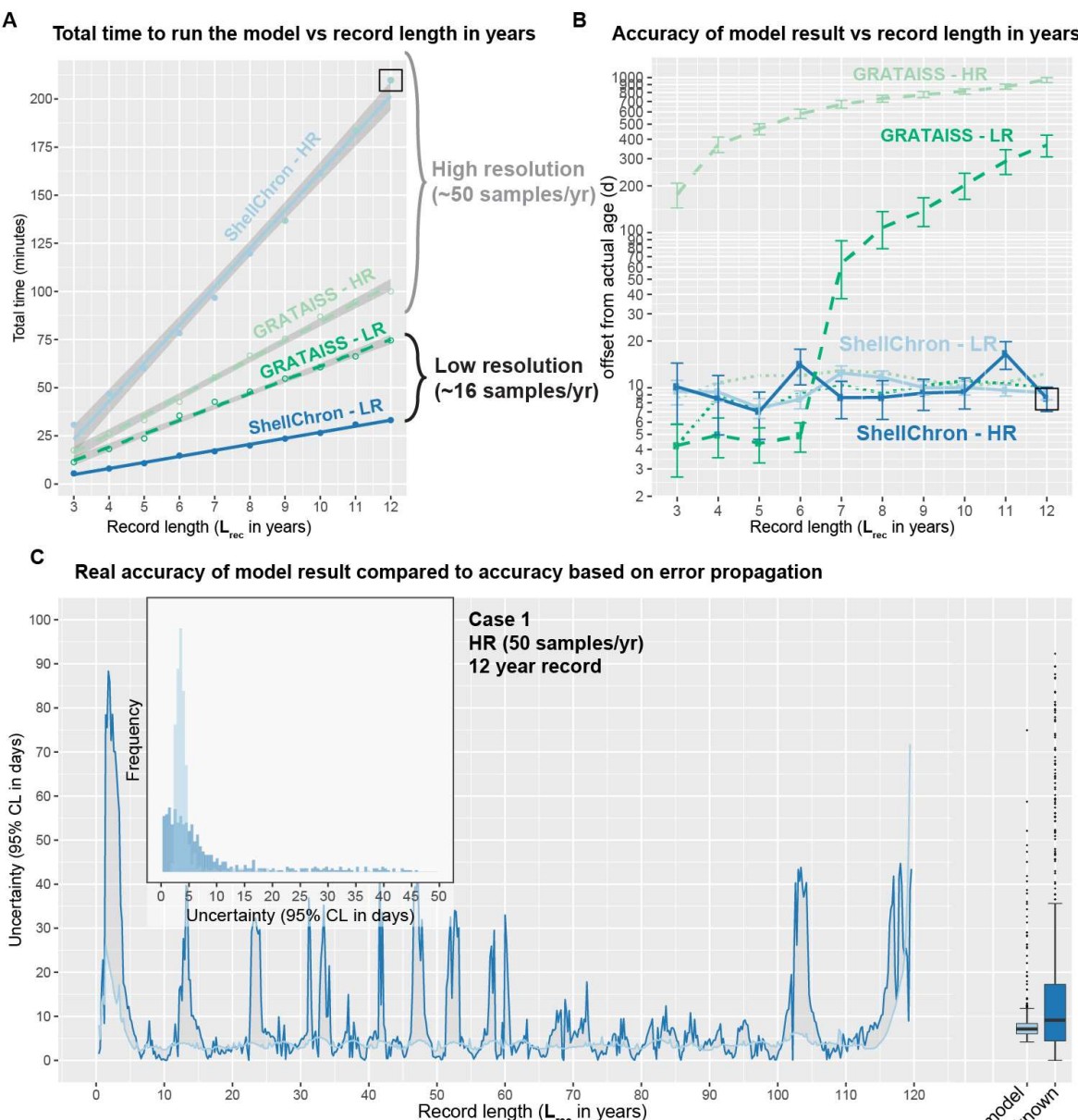


**Figure 8**: Overview of the result of timing ShellChron and the GRATAISS model on the same datasets (**A**), comparing the accuracies of both models (**B**) and comparing the accuracy as calculated by ShellChron with the known offset in the age model (**C**). In (**A**) and (**B**), low resolution datasets are plotted in dark blue (ShellChron) and dark green (GRATAISS), while high-resolution datasets plot in light blue (ShellChron) and light green (GRATAISS). Solid lines represent ShellChron and dashed lines show performance of the GRATAISS model. Green dotted lines in (**B**) show the accuracies of the GRATAISS model on a year-by-year basis (without accumulating error due to linking consecutive years) The black box in (**A**) and (**B**) highlights the dataset used in (**C**). In (**C**), dark blue lines, bars and boxplot indicate true offset of the model from the actual sample age, while light blue lines, bars and boxplot show the

542    accuracy of the model as calculated from the propagated errors on model and input data. Raw data is

543    provided in **SI11**.

**4.4 Modeling time**

The performance of both ShellChron and GRATAISS in terms of computation time linearly increases with the length of the record (in years; see **Fig. 8**, **Fig. S5** and **SI11**). Computation time of ShellChron on the high-resolution test dataset (50 samples/yr) increases very steeply with the length of the record in years (~20 minutes per additional year), while the low-resolution dataset (16 samples/yr) shows a slower increase (~3 minutes per additional year; **Fig. 5A**). This contrasts with GRATAISS, which requires only slightly more time on high-resolution data than on low-resolution datasets (~7 and ~10 minutes per additional year, respectively). The difference is explained by the sliding window approach applied in ShellChron, which requires more SCEUA optimization runs per year in high-resolution datasets than in low resolution datasets. When plotted against the number of calculation windows or samples in the dataset, running ShellChron on low-resolution and high-resolution datasets require a similar increase in computation time (~0.4 minutes, or 24 seconds, per additional sample/window; **Fig. S5**) under default SCEUA conditions. ShellChron outcompetes GRATAISS in terms of computation time in datasets with fewer than ~20 samples per year, even though more SCEUA optimizations are required.

A key computational improvement in ShellChron is the application of a sinusoidal regression before each SCEUA optimization to estimate the initial values of the modelled parameters (*sinreg* function; see **equation 9** and **Fig. 1** in **Model description**). Since carbonate archives are rarely sampled for stable isotope measurements above 20 samples per year (e.g. Goodwin et al., 2003; Schöne et al., 2005; Lough, 2010 and references therein), the disadvantage of a steep computational increase for very high-resolution archives is, in practice, a favorable tradeoff for the added control on model and measurement uncertainty and smoother inter-year transitions ShellChron offers in comparison to previous models. The similarity of ShellChron's accuracy in the low- and high-resolution datasets demonstrates its robustness across datasets with various sampling resolutions (see also **Table 2** and **Fig. 7**).

Longer computation times in GRATAISS result in slightly better accuracy on the modelled age compared to ShellChron on the scale of individual datapoints in low-resolution datasets (see **Fig. 8B**). However, this advantage is rapidly lost when records containing multiple years are considered (**Fig. 8B**). The advantage of the ShellChron model is its application of overlapping model windows, which smooth out the transitions between modelled years and eliminate accumulations of model inaccuracies when records grow longer. In addition, contrary to previous models, ShellChron does not rely on user-defined

year boundaries, which may introduce mismatches between subsequent years to be propagated
through the age model, even in ideal datasets such as **Case 1** (**Fig. 8B**; see also **Supplementary**
**Methods**). By comparison, the overall accuracy of ShellChron is much more stable within and between
datasets of different length, while rarely introducing offsets of more than a month. It must be noted here
that the cumulative, multi-year age uncertainty in the GRATAISS model (**Fig. 8B**) was calculated by
combining the results of consecutive growth years in the record, which the GRATAISS model models
separately, while avoiding age inversions and retaining the seasonal phase of the model results. This
procedure causes gaps in time to be introduced in the cumulative age modelled by GRATAISS
whenever the results of two consecutive, individually modelled growth years do not align, explaining the
sharp increases in age uncertainty of the GRATAISS model result (**Fig. 8B**). These cumulative
uncertainties are therefore not theoretically part of the model result (see year-by-year uncertainty in **Fig.**
**8B**) but are a necessary consequence of the way GRATAISS approximates growth years separately. If
only within-year inaccuracies are compared, GRATAISS results are roughly equally accurate as
ShellChron results (see dotted lines in **Fig. 8B**).
Where ShellChron considers the uncertainty on input parameters, this uncertainty is not considered in
most previous models (the MoGroFun model of Goodwin et al., 2003 being the exception). The added
uncertainty caused by input error is higher in less regular (sinusoidal) $\delta^{18}O_c$ records and in records with
lower sampling resolution, causing the uncertainties on GRATAISS reported here for the ideal, high-
resolution **Case 1** dataset to be over-optimistic. If ShellChron's model accuracy is insufficient, its
modular character allows the user to run the SCEUA algorithm to within more precise optimization
criteria by changing the model parameters (see **section 4.1**). However, this adaptation comes at a cost
of longer computation times.
The estimated uncertainty envelope (95% confidence interval) on the modelled age calculated by the
error propagation algorithm in ShellChron (4.7 ± 6.5 d) on average slightly underestimates the actual
offset between modelled age and known age in the **Case 1** record (9.3 ± 13.1 d; **Fig. 8C**). The
foremost difference between modelled and known uncertainty on the result is that the modelled
uncertainty yields a more smoothed record of uncertainty compared to the record of actual offset of the
model (**Fig. 8C**). ShellChron's uncertainty calculations are partly based on comparing overlapping
model windows, thereby smoothing out short term variations in model offset. The uncertainty of the
model result (both known and modelled) shows regular variability with a period of half a year (**Fig. 8C**).
Comparing this variability with the phase of the record (of which 6 years are plotted in **Fig. 6A**) reveals
that the uncertainty of the model is negatively correlated to the slope of the $\delta^{18}O_c$ record. This is
expected, because in parts of the record with extreme values in the $\delta^{18}O_c$ curve, the local age model
result is more sensitive to small changes in the sampling distance, caused either by uncertainty in the
model fit or propagated uncertainty on the sampling distance defined by the user (see discussion in
section 4.2). The slight seasonal variability in model accuracy in **Case 1** is also shown in **Fig. 6C** and
comprises a difference in uncertainty of up to 10 days depending on the time of year in which the
datapoint is found.

**5. Applications and discussion**


Its new features compared to previous age model routines make ShellChron a versatile package for
creating age models in a range of high-resolution paleoclimate records. The discussion above
demonstrates that ShellChron can reconstruct the age of individual $\delta^{18}O_c$ samples with monthly
precision. This level of precision is sufficient for accurate reconstructions of seasonality, defined as the
difference between warmest and coldest month (following USGS definitions; O'Donnell and Ignizio,
2012). While an improvement on this uncertainty could be of potential interest for ultra-high-resolution
paleoclimate studies (e.g. sub-daily variability, see Sano et al., 2012; Yan et al., 2020; de Winter et al.,
2020a), the increase in computation time and the sampling resolution such detailed age models demand
render age modelling from $\delta^{18}O_c$ records inefficient for this purpose (see **sections 4.1** and **4.4**). The
sampling resolution for high-resolution carbonate $\delta^{18}O_c$ records in the literature does not typically exceed
100 μm due to limitations in sampling acquisition (e.g. micromilling), which even in fast-growing archives
limits the resolution of these records to several days at best (see Gagan et al., 1994; Van Rampelbergh
et al., 2014; de Winter et al., 2020c). While in some archives, high-resolution (< 100 μm) trace element
records could be used to capture variability beyond this limit, the monthly age resolution of ShellChron
is sufficient for most typical high-resolution paleoclimate studies.
The ability to produce uninterrupted age models from multi-year records while considering both
variability in $\delta^{18}O_w$ and uncertainties on input parameters represent major advantages of ShellChron
over previous age modelling solutions. As a result, ShellChron can be applied on a wide range of
carbonate archives (see **Fig. 7** and **Table 2**). However, testing ShellChron on different records highlights
the limitations of the model inherited through its underlying assumptions. The most accurate model
results are obtained on records with minimal growth rate and $\delta^{18}O_w$ variability and a nearly sinusoidal
$\delta^{18}O_c$ record, such as tropical **coral** records (**Fig. 7A**; Gagan et al., 1994). In records where large
seasonal variability in growth rate and $\delta^{18}O_w$ does occur, such as in intertidal **oyster** shells, ShellChron's
accuracy slightly decreases, especially near growth hiatuses in the record (see **Fig. 7B**; Ullmann et al.,
2010). A worst-case scenario is represented by the **speleothem** record, which not only suffers from
much slower and more unpredictable growth rates and contains a comparatively small annual range in
$\delta^{18}O_c$, but it responds to $\delta^{18}O_w$ variability in drip water in the cave rather than temperature seasonality,
one of the assumptions underlying the current version of ShellChron (**Fig. 7C**; Vansteenberghe et al.,

2019). Despite these problems, ShellChron yields an age model that is remarkably accurate on an annual timescale, which is as good as, or better than, the best age model that can be obtained by applying layer counting on the most clearly laminated parts of the speleothem (e.g. Verheyden et al., 2006). It must be noted that, while the close fit between modelled $\delta^{18}O_c$ and **speleothem** $\delta^{18}O_c$ data ($\rho$ = 0.92; $\sigma$ = 0.08‰) is encouraging, a major reason for the model's success is the fact that the Proserpine speleothem used in this example is known to receive significantly seasonal (though not sinusoidal) drip water volumes and concentrations (Van Rampelbergh et al., 2014). Variability in drip water properties and cave temperatures are known to differ strongly between cave systems (Fairchild et al., 2006; Lachniet, 2009). For ShellChron (or any other $\delta^{18}O_c$-based age model) to work reliably in speleothem records, consistent seasonal variability in either temperature or $\delta^{18}O_w$ should be demonstrated to significantly influence the $\delta^{18}O_c$ variability in the record. In practice, these constraints make ShellChron applicable in speleothems for which the cave environment varies in response to the seasonal cycle, such as localities overlain by thin epikarst, well-ventilated caves or speleothems situated close to the cave entrance (Verheyden et al., 2006; Feng et al., 2013; Baker et al., 2021).

ShellChron's ability to model multi-year records with smooth transitions between the years does not compromise the accuracy of its age determination on the seasonal scale (e.g. **Fig. 6** and **7**). Many paleoclimatology studies investigating the seasonal cycle rely on stacking of seasonal variability relative to the annual cycle, thereby combining seasonal information from multiple years to obtain a precise reconstruction of seasonal variability in the past (e.g. de Winter et al., 2018; Judd et al., 2019; Tierney et al., 2020). While this can be achieved using age models of individual years (e.g. Judd et al., 2018), seasonally resolved archives dated using ShellChron can also be stacked along a common seasonal axis while retaining information about the multi-annual record allowing, for example, comparison between consecutive years dated using the same age model including uncertainty on the age determination.

The difficulty of applying age model routines on speleothem records highlights one of the main advantages of ShellChron over pre-existing age model routines, namely its modular character. Since $\delta^{18}O_c$ records from some carbonate archives, such as speleothems, cannot be described by the standard combination of temperature and growth rate sinusoids on which ShellChron is based (in its current version), the possibility to adapt the "building block" functions used to approximate these $\delta^{18}O_c$

records (*d18O_model*, *temperature_curve* and *growth_rate_curve*; see **Fig. 1**) while leaving the core
structure of ShellChron intact greatly augments the versatility of the model. The freedom to adapt the
building blocks used to approximate the $\delta^{18}O_c$ record theoretically enables ShellChron to model sub-
annual age-distance relationships in any record if the seasonal variability in the variables used to model
the input data are predictable and can be represented by a function. For example, since speleothem
$\delta^{18}O_c$ records often depend on variability in the $\delta^{18}O_w$ value of the drip water, a function describing this
variability through the year can replace the *temperature_curve* function to create more accurate sub-
annual age models for speleothems (e.g. Mattey et al., 2008; Lachniet, 2009; Van Rampelbergh et al.,
2014). Similarly, the *growth_rate_curve* function can be modified in case the default skewed sinusoid
does not accurately describe the extension rate of the record under study, and the *d18O_model* function
can be adapted to feature the most fitting $\delta^{18}O_c$-temperature or $\delta^{18}O_c$-$\delta^{18}O_w$ relationship. Note that the
flexibility of this approach is limited by the expression of the annual cycle in the $\delta^{18}O_c$ record. The $\delta^{18}O_c$-
based dating approach in ShellChron will therefore have more trouble dating records in which the annual
$\delta^{18}O_c$ variability is severely dampened, such as speleothems in deeper cave systems (e.g.
Vansteenberge et al., 2016), or in which annual $\delta^{18}O_c$ variability is not sinusoidal, such as tropical
records with bimodal temperature or precipitation seasonality (Knoben et al., 2018).
Flexibility in the definition of "building block" functions used to approximate the input data paves the way
for future application beyond carbonate $\delta^{18}O_c$ records. The seasonal variability in $\delta^{18}O$ in some ice cores
can be approximated by a stable and unbiased temperature relationship (van Ommen and Morgan,
1997). ShellChron can therefore be modified to date sub-annual samples in these ice core records and
reconstruct seasonal variability in the high latitudes through the Quaternary. Similarly, inter-annual $\delta^{18}O$
variability in tree ring records are demonstrated to record variability in precipitation through the year,
and this variability can be modelled to improve sub-annual age models in these records (Xu et al., 2016).
More generally, the field of dendrochemistry has recently developed additional chemical proxies for
seasonality (e.g. trace element concentrations), which can be measured on smaller sample volumes
(and thus greater resolution) to obtain ultra-high-resolution records on which (sub-annual) dating can be
based (e.g. Poussart et al., 2006; Superville et al., 2017). A similar development has taken place in the
study of carbonate bio-archives such as corals and mollusks, of which some show strong, predictable
seasonal variability in trace elements (e.g. Mg/Ca and Sr/Ca ratios) which can be used to accurately
date these records (de Villiers et al., 1995; Sosdian et al., 2006; Durham et al., 2017; de Winter et al.,
2021b). Minor changes in the "building block" functions using empirical transfer functions for these trace
element records will enable ShellChron to capitalize on these relationships and reconstruct sub-annual
growth rates with improved precision due to the higher precision with which these proxies can be
measured compared to $\delta^{18}O_c$ records. Finally, the application of ShellChron for age model construction
is not necessarily limited to the seasonal cycle, as other major cycles in climate (e.g. tidal, diurnal or
Milankovitch cycles) leave similar marks on climate records and can thus be used as basis for age
modelling (e.g. Sano et al., 2012; Huyghe et al., 2019; de Winter et al., 2020a; Sinnesael et al., 2020).
It must be noted that, since ShellChron was developed for modeling based on annual periodicity,
applying it on other timescales would require more thorough adaptation of the model code than merely
adapting the "building block" functions to support additional proxy systems.
While age reconstructions are the main aim of ShellChron, the model also yields information about the
temperature and growth rate parameters used in each simulation window to approximate the local $\delta^{18}O_c$
curve (see *parameter matrix* in **Fig. 1** and **SI6**). These parameters hold key information about the
response of the archive to seasonal changes in the environment, such as the season of growth,
relationships between growth rate and temperature and the temperature range that is recorded.
Combining these parameters with records of influential environmental variables such as seawater
chlorophyl concentration or local precipitation patterns yields information about the response of the
climate archive to environmental variables, in addition to the climate or environmental change it records.
Study examples include the relationship between growth rate of marine calcifies and phytoplankton
abundance or the correlation between precipitation patterns and chemical variability in speleothems.
While such discussion is beyond the scope of this work, examples of parameter distributions are
provided in **SI5**, and the application of modelled growth rate parameters in bivalve sclerochronology is
discussed in more detail in Judd et al. (2018). Note that the sliding window approach of ShellChron
produces records of changing temperature and growth rate parameters at the scale of individual
samples (albeit smoothed by the sliding window approach) rather than annually, as in Judd et al. (2018).

**6. Conclusions**
ShellChron offers a novel, open-source solution to the problem of dating carbonate archives for high-
resolution paleoclimate reconstruction on a sub-annual scale. Based on critical evaluation of previous
age models, building on their strengths while attempting to minimize their weaknesses, ShellChron
provides continuous age models based on $\delta^{18}O_c$-profiles in these archives with monthly accuracy,
considering the uncertainties associated with both the model itself and the input data. The monthly
accuracy of the model, as tested on a range of virtual and natural datasets, enables its application for
age determination in studies of seasonal climate and environmental variability. Higher accuracies can
be reached at the cost of longer computation times by adapting the model parameters, but age
determinations far beyond the monthly scale are unlikely to be feasible considering the limitations on
sampling resolution and measurement uncertainties on $\delta^{18}O_c$ records. ShellChron's computation times
on datasets with sampling resolutions typical for the paleoclimatology field (up to 20 samples/yr) remain
practical and comparable to previous model solutions, despite adding several features that improve the
versatility and interpretation of model results. Its modular design allows ShellChron to be adapted to
different situations with comparative ease. It thereby functions as a platform for age-distance modelling
on a wide range of climate and environmental archives and is not limited in its application to the $\delta^{18}O_c$
proxy, the carbonate substrate or even to the annual cycle, as long as the relationship between the
proxy and the extension rate of the archive on a given time scale can be parameterized. Future
improvements will capitalize on this variability, expanding ShellChron beyond its current dependency on
the $\delta^{18}O_c$-temperature relationship in carbonates. Members of the high-resolution paleoclimate
community are invited to contribute to this effort by adapting the model for their purpose.

**Code availability**
ShellChron is worked out into a fully functioning package for the open-source computational language
R (version 3.5.0 or later; R Core Team, 2020). The most recent full version (v0.4.0) of the ShellChron
passed the code review of the Comprehensive R Archive Network (CRAN) and is freely available for
download as an R package on the CRAN server (see https://CRAN.R-project.org/package=ShellChron).
The CRAN server entry also includes detailed line-by-line documentation of the code and working
examples for every function. In addition, the latest development version of ShellChron is available on
GitHub (https://github.com/nielsjdewinter/ShellChron). Those interested in adapting ShellChron for their
research purposes are invited to do so there. Code and documentation, together with all supplementary
files belonging to this study, are also available on the open-source online repository Zenodo
(http://doi.org/10.5281/zenodo.4288344).

**Author contribution**
NJW designed the study, wrote the model script, carried out the test calculations and wrote the
manuscript.

**Competing interests**
There were no competing interests to declare.

**Acknowledgements**
This research project is part of the UNBIAS project funded by the European Commission through a
Marie Curie Individual Fellowship (MSCA-IF; grant number: 843011) and the Flemish Research Council
(FWO; junior postdoc grant, project number: 12ZB220N). Thanks go to Emily Judd for discussions about
the workings of the Judd et al. (2018) model and its potential adaptation beyond aragonitic mollusk
shells. High-resolution temperature and salinity data from the NIOZ jetty which underlie the **Texel**
dataset and the noise added to the idealized **Case 1** dataset were kindly provided by Eric Wagemaakers
and Sonja van Leeuwen (Royal Dutch Institute for Sea Research, the Netherlands). The $\delta^{18}O_c$ data
series from the *Crassostrea gigas* (**oyster**) and Proserpine stalagmite (**speleothem**) were generously
provided by dr. Clemens V. Ullmann (University of Exeter, UK) and dr. Stef Vansteenberge (Vrije
Universiteit Brussel, Belgium), respectively. Raw data from the *Porites lutea* **coral** dataset were obtained
with help of the WebPlotDigitizer (https://automeris.io/WebPlotDigitizer/) developed by Ankit Rohatgi.
Preparation of the ShellChron model into an R package would not have been possible without the helpful
instructions by Fong Chun Chan (https://tinyheero.github.io/jekyll/update/2015/07/26/making-your-first-
R-package.html), Hilary Parker (https://hilaryparker.com/2014/04/29/writing-an-r-package-from-
scratch/) and Hadley Wickham (https://r-pkgs.org/release.html) as well as the insightful and inspiring
discussions on R coding and statistics with Ilja Kocken (Utrecht University). In addition, distribution of
the code in an organized way was made possible thanks to Git (https://git-scm.com/) and Github
(https://github.com/) and the R Project Team (https://www.r-project.org/), with special thanks to Uwe
Ligges (University of Dortmund, Germany) and Gregor Seyer (University of Vienna, Austria) for their
comments on initial submissions of the package to the CRAN database. Thanks go to William A. Huber
(https://www.analysisandinference.com/team/william-a-huber-phd) for providing a practical general
solution to the peak identification problem in the *cumulative_day* function (see *peakid* function and
https://rpubs.com/mengxu/peak_detection).

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
