# Peer review of "ShellChron 0.4.0: A new tool for constructing chronologies in accretionary carbonate archives"

_Geoscientific Model Development, 2020_

## Referee Comment (RC1)

In *ShellChron 0.2.8: A new tool for constructing chronologies in accretionary carbonate archives from stable oxygen isotope profiles*, de Winter presents a model for temporally aligning sclerochronologic data. The approach expands on the growth rate model of Judd et al. (2018), with three notable improvements: (1) a sliding window approach that allows for more continuous age transformation of multi-year records, (2) a modular code design that increases usability and flexibility of application (e.g., user-specified proxy-to-temperature transfer function), and (3) propagation of error and uncertainty.

In premise, the approach offers an innovative solution to a complex problem and promises to elegantly addresses the key limitations of previous growth models. However, after reading the manuscript several times and going through the supplement, I am still left with several questions about the practical application of the approach, including but not limited to:

(1) **Attenuated amplitude of the modelled isotope data**: Why do the modelled curves seem to consistently underestimate the known isotopic range (e.g., **Fig 3**)? The manuscript explains that each window is designed to include at least one full year of growth (L238), meaning that at least one (if not both) seasonal extremes should be included in any given window. Additionally, the stated benefit of fitting a (linearized) sinusoid to the data in the window (in the depth domain) prior to iterating the temperature and growth rate functions is to set realistic starting parameters and bounds (L267-272). Given this, it seems the modelled isotope curve of each window should accurately capture the seasonal range of values.

This point is illustrated in **Fig. S1**, which shows the known and weighted mean modelled isotope curves for the Texel high resolution scenario (**Fig. S1A**) and the residuals (**Fig. S1B**). Consistent with **Fig. 3**, the residuals are normally distributed and center at 0‰; however, they exhibit a non-random distribution, paralleling the seasonal signal – a direct result of systematically underestimating seasonal extremes. This is important as it not only detracts from the potential climatological and biological insights from the temperature and growth rate functions, it also leads to spurious intra-annual temporal alignment of the data.

(2) **Number of growing days**: In looking through the supplement for the Texel high resolution scenario, more than 35% of the windows predict that the duration of growth captured by the isotope data in each window is less than 7 days (as per '*Day_of_year_raw.csv*'). Again, if each window spans at least one year and realistic bounds have been set for the temperature and growth rate functions, then why is growth condensed into such a short duration?

(3) **Shape of the modelled isotope curves in the depth and time domains**: Why do some modelled isotope curves exhibit high frequency variability (e.g., Window 3 of the Texel high resolution scenario; **Fig. S2**)? My understanding is that the modelled curves derive from the sinusoidal temperature function and the skewed sinusoidal growth rate function, both of which exhibit monotonic increases or decreases between their peaks and troughs. Are these wiggles a function of the weighting? If so, why is it not consistent across all windows? Also, as alluded to above, many of the windowed curves exhibit long flat lines at their start and/or end, indicating that the temperature and growth rate function bounds might be impeding a better fit.

I suspect there might be an error in the Case 1 and Texel examples, as some of the concerns addressed above don't appear in the Coral, Oyster, and/or Speleothem examples. (Could the upper amplitude bound accidentally been set at $T_{amp}/2$ in these examples?) However, it is difficult to (a) ensure that I fully understand the methodology and (b) evaluate the fidelity and applicability of the approach while these questions linger.

**Other general comments**

- The manuscript would benefit from a detailed guided example. It is difficult to clearly and concisely convey complex mathematical model. Rather than presenting the Case 1 and Texel examples (which ultimately don't add much insight that can't be gleaned from the real-world examples), I would suggest creating a shorter (<5 yearlong) virtual dataset and stepping the reader through 2 or 3 window examples, showing the temperature and growth rate functions for each window and comparing those to the known values.

- It would be useful to address how the moving window approach is impacted by changes to the number of samples per year change and/or interannual extension rate. In practice, it can sometimes be difficult to sample at a constant rate and/or samples are sometimes lost or too small to run. How would the moving window approach adapt to an example where the first year had 15 samples and the second had only 8? Similarly, how are the results impacted if there is a large change to the interannual extension rate (as is observed in some low latitude bivalves)?
- Can you please clarify whether the period of the window (in the time domain) is defined during the linearized sinusoidal fit or if it is fixed at 365 days?

**Line-by-line comments**

L22: This notation is not defined until L70; I'd suggest rephrasing to 'oxygen isotope records'

L39: Change 'time' to 'temporal'

L54: Delete 'of'

L54: Change 'greenhouse warming' to 'anthropogenic warming' or something of the like – we're still a far way off from a true ice-free greenhouse climate

L58: Change 'by' to 'at'

L71: Change 'by' to 'at'

L73: 'one being dominant over the other' – I'm not sure I completely agree; it can be quite difficult to disentangle these signals with, for example, low latitude, shallow bivalves and corals that are subject to seasonal precipitation; perhaps revise to 'with one generally being dominant over the other'

L91: Change 'causes them to rely on' to 'require'

L109: Define SCE-UA (Shuffled Complex Evolution model developed at the University of Arizona)

L112: Sample trajectories are referred to as "depths" in the text of the manuscript but "lengths" in the figures (I'd suggest opting for "sample distance", as that term is more widely applicable ("depth" is an odd term in reference to bivalves)

L138: Rephrase 'the question which'

L138 – 154: The original foraminifera references should be Spero et al. (1997) and Zeebe (1999). However, I'd contend that these studies do not suggest that foraminifera are precipitated out of equilibrium with their environment, but rather that additional environmental variables pertaining to the carbonate system (e.g., pH) contribute to oxygen isotope value of carbonate. This discussion is important (!) but highly nuanced, and I'm not sure this is the appropriate platform to address it. I'd suggest removing this section or moving it to the supplement; the important part of this paragraph as it pertains to the work presented in the paper is that ShellChron permits the user to define their desired transfer function.

L162-163: Rephrase 'temperature evolution'

L168: Change 'priory' to 'priori'

L206 (and all other instances): I don't know that 'depth' is the correct word choice; bivalves aren't generally discussed in terms of a depth domain, but instead a sampling 'distance' along a growth axis

L211,212: Change both instances of 'were' to 'are' for consistency of tense throughout the paragraph

L316: delete 'in the'

**New references cited**

Spero, H.J. et al. (1997). Effect of seawater carbonate concentration on foraminiferal carbon and oxygen isotopes. *Nature*, *390*(6659). https://doi.org/10.1038/37333.

Zeebe, R.E. (1999). An explanation of the effect of seawater carbonate concentration on foraminiferal oxygen isotopes. *Geochim. Cosmochim. Acta,* (63). https://doi.org/10.1016/S0016-7037(99)00091-5.

[Figure]

**Figure S1:** (**A**) Known and modelled isotope profiles from the Texel high resolution example. Known values come directly from the supplemental file '*SI8_Texel_data.xlsx*' and mean weighted modelled values come from the supplemental file '*d18O_model_results.csv*'. The modeled isotope profile consistently underestimates the seasonal extremes. (**B**) Model residuals (known minus modelled values) plotted as a function of sampling distance. Though the residuals are normally distributed around 0, as shown in the histogram of **Fig. 3**, they exhibit a non-random, highly seasonal pattern suggesting that there is a systematic misfit of the data.

[Figure]

**Figure S2**: First 12 mm of the Texel example, with the modelled isotope curve from Window 3 of the high resolution scenario overlain. Known values again come directly from the supplemental file '*SI8_Texel_data.xlsx*' and the Window 3 values come from the supplemental file '*modelled_d18O_raw.csv*'. The modelled curve exhibits high frequency variability (i.e., wiggles), inconsistent with a smooth sinusoidal fit.

---

## Author Comment (AC1)

Dear Geoscientific Model Development Editorial Board, Dear reviewers,

Let me start by thanking both reviewers for their critical and constructive feedback on my manuscript. Both reviewers are quite critical on the ShellChron approach and raise valid concerns about its performance. Below, I will first attempt to summarize the main concerns raised by the reviewers and reply in general. Later, I will provide a point-by-point reply to the comments posed by the reviewers.

Firstly, I do not agree with Reviewer #2 that the model is oversophisticated and that it has no advantage over linear interpolation between seasonal extremes. As stated by Reviewer #1, there exist many seasonal archives with strong seasonal variability in extension rate (e.g. low-latitude bivalve shells). In most cases, approximating the seasonal curve through sinusoidal modelling, as is done by both ShellChron and the model by Judd et al. (2018), yields superior results over linear interpolation between seasonal extremes.

That said, the reviewers highlight several key characteristics in ShellChron's output which indicate either a lack of clarity in the explanation of the calculations or sub-optimal implementation of the model. The reviewers give helpful suggestions on how the explanation can be improved, for example by including a step-by-step guide through a simpler (lower resolution) example and clearer explanation of how the parameters of ShellChron (e.g. days per year and length of modelling windows) are defined. These issues can be solved through an update of the manuscript text.

Issues related to the implementation of the model include the underestimation of the seasonal temperature amplitude highlighted by both reviewers, the occurrence of "flat lines" in individual modelling windows and the occurrence of high-frequency variability in the modelled temperature and growth curve after weighing and combining results from multiple modelling windows. Issues pertaining to the shape and fit of individual modelling windows relate to the way the SCEUA algorithm is implemented, and how its starting parameters are set in ShellChron (see comment by Reviewer #1). The overall match of the model to the multi-year $\delta^{18}O$ curve depends on the combination of consecutive growth years and the recognition of year transitions in the model. Improving these aspects of ShellChron requires updates to the code, after which the examples in the manuscript have to be reproduced.

I will reply in detail to the comments raised by both reviewers below to highlight how I plan to respond to the review comments and improve the ShellChron code and the manuscript according to their suggestions.

**Reviewer #1**

In *ShellChron 0.2.8: A new tool for constructing chronologies in accretionary carbonate archives from stable oxygen isotope profiles*, de Winter presents a model for temporally aligning sclerochronologic data. The approach expands on the growth rate model of Judd et al. (2018), with three notable improvements: (1) a sliding window approach that allows for more continuous age transformation of multi-year records, (2) a modular code design that increases usability and flexibility of application (e.g., user-specified proxy-to-temperature transfer function), and (3) propagation of error and uncertainty.

In premise, the approach offers an innovative solution to a complex problem and promises to elegantly addresses the key limitations of previous growth models. However, after reading the manuscript several

times and going through the supplement, I am still left with several questions about the practical application of the approach, including but not limited to:

(1) **Attenuated amplitude of the modelled isotope data**: Why do the modelled curves seem to consistently underestimate the known isotopic range (e.g., **Fig 3**)? The manuscript explains that each window is designed to include at least one full year of growth (L238), meaning that at least one (if not both) seasonal extremes should be included in any given window. Additionally, the stated benefit of fitting a (linearized) sinusoid to the data in the window (in the depth domain) prior to iterating the temperature and growth rate functions is to set realistic starting parameters and bounds (L267-272). Given this, it seems the modelled isotope curve of each window should accurately capture the seasonal range of values. This point is illustrated in **Fig. S1**, which shows the known and weighted mean modelled isotope curves for the Texel high resolution scenario (**Fig. S1A**) and the residuals (**Fig. S1B**). Consistent with **Fig. 3**, the residuals are normally distributed and center at 0‰; however, they exhibit a non-random distribution, paralleling the seasonal signal – a direct result of systematically underestimating seasonal extremes. This is important as it not only detracts from the potential climatological and biological insights from the temperature and growth rate functions, it also leads to spurious intra-annual temporal alignment of the data.

This is a valid point. Indeed, ShellChron systematically underestimates the total seasonal amplitude in the data. The reason for this lies in the (weighed) averaging of overlapping model windows. As illustrated in **Fig. 3**, individual model windows are not very sensitive to single datapoints, causing a slight smoothing of the seasonal $\delta^{18}O$ curve. This effect is exacerbated by the weighing of the $\delta^{18}O$ data within model windows, which causes datapoints on the edge of the model window to have lesser influence on the shape of the modelled curve. This weighing is implemented to prevent outliers in the data to influence the model outcome. The downside of this design choice is that individual datapoints in the extreme seasons (especially clear in the Texel case) fail to draw the simulation to the recorded maxima and minima in $\delta^{18}O$.

Another factor that may contribute to the poor fit quality of some model windows might be the freedom given to the SCEUA algorithm. Currently, the algorithm is set to stop optimization when function values ($\delta^{18}O$) and model parameters do not change by more than 0.01% between consecutive model runs ("pcento" and "peps" parameters in the "run_model.r" function), and the model is limited to 10000 function evaluations ("maxn" in "run_model.r") to limit processing time. Since large $\delta^{18}O$ datasets and overlapping modelling windows cause full ShellChron runs to already take over one hour on a modern personal computer in the current setup (as noted by Reviewer #2), limiting processing time is valuable. The sinusoidal fit was implemented to help the algorithm to find better starting values and cause the SCEUA optimization to converge within the set parameters. However, I concur that the advantage of this sinusoidal fit may have been overestimated. I will implement an option in ShellChron for the user to set the constraints on the optimization algorithm manually and to switch the sinusoidal fit off. To evaluate if the limitations placed on the SCEUA algorithm and/or the use of sinusoidal fitting cause underestimation of the seasonal $\delta^{18}O$ amplitude, I will test a range of SCEUA settings with and without sinusoidal fit on a small test dataset and describe the results of these tests in the manuscript.

While the underestimation of the seasonal amplitude in $\delta^{18}O$ data does indeed dampen the seasonal cycle of the modelled temperature curve, it should not negatively affect the modelled age-depth relationship (as was recognized by Reviewer #2, see below). To test this, I propose to include details on the residuals of the age-depth model for the test cases (Case 1 and Texel, see **Fig. 3**) into the manuscript and discuss any systematic (seasonal) offsets in the age alignment of the $\delta^{18}O$ data that may result from underestimation of the seasonal $\delta^{18}O$ amplitude.

(2) **Number of growing days**: In looking through the supplement for the Texel high resolution scenario, more than 35% of the windows predict that the duration of growth captured by the isotope data in each window is less than 7 days (as per '*Day_of_year_raw.csv*'). Again, if each window spans at least one year and realistic bounds have been set for the temperature and growth rate functions, then why is growth condensed into such a short duration?

The reviewer refers to the "flat lines" in some individual modelling windows, where ShellChron fails to properly approximate the seasonal $\delta^{18}O$ curve (see following comment). This issue is related to the previous point raised by the reviewer, and likely results from either a poor choice of initial modelling conditions through the sinusoidal fit, or from overly restrictive parameters of the SCEUA algorithm. If not properly recognized and removed, these flat model results may be partially responsible for dampening of the seasonal $\delta^{18}O$ curve by the model, because they are included in the average of the modelled $\delta^{18}O$ curve. As stated above, I intend to test the effect of the sinusoidal fit and limits on the SCEUA algorithm to assess whether these parameters compromise the model fit. In addition, I will re-evaluate the code to make sure that any remaining "flat" parts of individual modelling windows do not contribute to the average.

(3) **Shape of the modelled isotope curves in the depth and time domains**: Why do some modelled isotope curves exhibit high frequency variability (e.g., Window 3 of the Texel high resolution scenario; **Fig. S2**)? My understanding is that the modelled curves derive from the sinusoidal temperature function and the skewed sinusoidal growth rate function, both of which exhibit monotonic increases or decreases between their peaks and troughs. Are these wiggles a function of the weighting? If so, why is it not consistent across all windows? Also, as alluded to above, many of the windowed curves exhibit long flat lines at their start and/or end, indicating that the temperature and growth rate function bounds might be impeding a better fit. I suspect there might be an error in the Case 1 and Texel examples, as some of the concerns addressed above don't appear in the Coral, Oyster, and/or Speleothem examples. (Could the upper amplitude bound accidentally been set at $T_{amp}/2$ in these examples?) However, it is difficult to (a) ensure that I fully understand the methodology and (b) evaluate the fidelity and applicability of the approach while these questions linger.

Individual model simulation ("windows") are constrained by the two sinusoids and will therefore not exhibit high-frequency variability. On checking the supplementary file **SI7** from which the data in the reviewer's **Fig. S2** is obtained I discovered that the high-frequency variability only occurs in modelled $\delta^{18}O$ data from Window 3, and only in the high-resolution Texel dataset. None of the modelling window results from any of the tested datasets (see **SI7** and **SI8**) exhibit this variability, and I am afraid the incompatible data in Window 3 of the high-resolution test on the Texel data results from a copying error. My apologies for the inconvenience. I invite the reviewers to verify that no high-resolution variability occurs in the other window simulations, as indeed such variability cannot result from the combination of two sinusoids. I will provide a clean export of the raw results of the model on the high-resolution Texel dataset after addressing the issues raised above.

**Other general comments**

• The manuscript would benefit from a detailed guided example. It is difficult to clearly and concisely convey complex mathematical model. Rather than presenting the Case 1 and Texel examples (which ultimately don't add much insight that can't be gleaned from the real-world examples), I would suggest

creating a shorter (<5 yearlong) virtual dataset and stepping the reader through 2 or 3 window examples, showing the temperature and growth rate functions for each window and comparing those to the known values.

This is a great suggestion, and I am happy to implement it in a revised version of the manuscript. From the review comments it is clear that the manuscript as well as the ShellChron package would benefit from a step-by-step guided example through the model using a realistic $\delta^{18}$O record.

• It would be useful to address how the moving window approach is impacted by changes to the number of samples per year change and/or interannual extension rate. In practice, it can sometimes be difficult to sample at a constant rate and/or samples are sometimes lost or too small to run. How would the moving window approach adapt to an example where the first year had 15 samples and the second had only 8? Similarly, how are the results impacted if there is a large change to the interannual extension rate (as is observed in some low latitude bivalves)?

A good suggestion, and I would happily implement it by testing ShellChron on a more realistic (lower sampling resolution) virtual dataset with varying extension rate. Note that the Texel dataset currently implemented already features a decreasing extension rate, but I concur that it may be more interesting to test a more realistic $\delta^{18}$O record with lower sampling resolution in which this decrease occurs.

• Can you please clarify whether the period of the window (in the time domain) is defined during the linearized sinusoidal fit or if it is fixed at 365 days?

The period of one year, as defined by the model outcome, is set at 365 days (although it is possible for the user to override this default through the "T_per" parameter). Window length is defined based on linear interpolation between the user provided YEARMARKERS such that each window contains at least one growth year. The length of a full window can therefore be longer than 365 days, as the length of one year in sampling direction is determined by the model outcome based on the $\delta^{18}$O data. An initial value for the period in sampling direction is found through the sinusoidal fit before the SCEUA algorithm optimizes the growth rate and temperature sinusoids. These points will be clarified in the revised manuscript.

**Line-by-line comments**

L22: This notation is not defined until L70; I'd suggest rephrasing to 'oxygen isotope records'

I will rephrase this accordingly in the Abstract.

L39: Change 'time' to 'temporal'

I will rephase to "temporal"

L54: Delete 'of'

"of" will be deleted

L54: Change 'greenhouse warming' to 'anthropogenic warming' or something of the like – we're still a far way off from a true ice-free greenhouse climate

Agreed, I did not want to suggest that we are in a greenhouse state, rather refer to "warming by greenhouse gases". I agree that "anthropogenic (global) warming" is better and will rephrase accordingly.

L58: Change 'by' to 'at'

"by" will be rephrased to "at"

L71: Change 'by' to 'at'

"by" will be rephrased to "at"

L73: 'one being dominant over the other' – I'm not sure I completely agree; it can be quite difficult to disentangle these signals with, for example, low latitude, shallow bivalves and corals that are subject to seasonal precipitation; perhaps revise to 'with one generally being dominant over the other'

Agreed, I will rephrase to "with one generally being dominant over the other".

L91: Change 'causes them to rely on' to 'require'

I will rephrase to "requires"

L109: Define SCE-UA (Shuffled Complex Evolution model developed at the University of Arizona)

The abbreviation will be defined here.

L112: Sample trajectories are referred to as "depths" in the text of the manuscript but "lengths" in the figures (I'd suggest opting for "sample distance", as that term is more widely applicable ("depth" is an odd term in reference to bivalves)

This is a good point, and I will refer to "sample distance" or "distance in sampling direction" throughout the manuscript.

L138: Rephrase 'the question which'

I will rephrase to "Which carbonates are…"

L138 – 154: The original foraminifera references should be Spero et al. (1997) and Zeebe (1999). However, I'd contend that these studies do not suggest that foraminifera are precipitated out of equilibrium with their environment, but rather that additional environmental variables pertaining to the carbonate system (e.g., pH) contribute to oxygen isotope value of carbonate. This discussion is important (!) but highly nuanced, and I'm not sure this is the appropriate platform to address it. I'd suggest removing this section or moving it to the supplement; the important part of this paragraph as it pertains to the work presented in the paper is that ShellChron permits the user to define their desired transfer function.

I agree that this is not the place to discuss the equilibrium fractionation of foraminifera or the effects of environmental parameters on their $\delta^{18}O$ value. Therefore, this paragraph will be shortened, focusing on the importance of ShellChron's flexibility in choosing different transfer functions for the $\delta^{18}O$-temperature relationship.

L162-163: Rephrase 'temperature evolution'

This will be rephrased to "the evolution of the calcification temperature"

L168: Change 'priory' to 'priori'

This will be rephrased accordingly.

L206 (and all other instances): I don't know that 'depth' is the correct word choice; bivalves aren't generallydiscussed in terms of a depth domain, but instead a sampling 'distance' along a growth axis

See above, I agree and will refer to "distance (in sampling direction)" throughout the manuscript.

L211,212: Change both instances of 'were' to 'are' for consistency of tense throughout the paragraph

This will be rephrased for consistency.

L316: delete 'in the'

This will be rephrased to "Wadden Sea (North Netherlands; Texel; see details in…

**Reviewer #2**

The paper presents the details of ShellChron, an R package that calculates the inner chronology of seasonally resolved oxygen isotopic records such as those obtained from corals, mollusks or some speleothems. ShellChron does so by calculating a d18O curve combining an assumed sinusoidal temperature curve, a quasi-sinusoidal growth curve, and a paleo-temperature equation, fit onto the d18O record in moving windows. The performance of the model is evaluated using synthetic and real records.

While I acknowledge the importance of the inner chronology issue in the study of seasonal scale processes from paleoclimate records, ShellChron seems like a heavy oversophisticated solution (several hours of calculations per depth-age model) with no clear advantage compared to simple techniques like matching the d18O record with local temperature curve or a simple linear interpolation between seasonal isotopic extrema used as anchor points.

I agree that, in modern cases where local temperature and salinity curves are available, a modelling approach like ShellChron or the model in Judd et al. (2018) is not necessary, because the $\delta^{18}O$ data can be aligned to those known temperature and salinity curves. However, in cases where "true" environmental data are not available, modelling the seasonality curve has clear advantages over simple linear interpolation between seasonal extremes. As discussed in lines 82-97, in most shallow marine carbonate archives both temperature and growth rate describe quasi-sinusoidal patterns. Linear interpolation between maxima and minima (assuming constant growth rates in between) cannot approximate the growth rate in these records as well as a sinusoidal curve (see discussion in Judd et al. 2018). Therefore, in absence of "true" temperature and salinity records (i.e. in fossil archives), the sinusoidal modelling approach is a much better assumption for age-modelling than linear interpolation.

The author interprets too positively the test results. In my view, the tests indicate low performance in terms of dating precision despite the high quality of the records (synthetic or real), which questions the usefulness of the whole package.

In my discussion in relation to Figures 3-5, I honestly discuss the uncertainties of ShellChron in terms of reproducibility of the model result (precision) as well as agreement with the "true" age-distance relationship of the archive (accuracy). It is true that the ages found by ShellChron tend to have uncertainties of ±30 days or higher (see e.g. lines 345-361), but the advantage of ShellChron is that this uncertainty is known and quantifiable. In previous models, and especially when assuming constant growth between seasonal extremes (see above), this uncertainty is unknown. Furthermore, uncertainties calculated by ShellChron uniquely consider uncertainties in both the $\delta^{18}O$ measurement

and the sample position, making these uncertainties seem higher than those in other models. If the reviewer believes the discussion of ShellChron's uncertainty is insufficient, I invite any comments indicating where the results of ShellChron are interpreted too positively and how these improvements could be made.

- It is said in the text that a seasonally varying water d18O curve can be included but it is not clear how to do so in practice.

This is a valid point. In fact, the $\delta^{18}O$ cure can be provided as a vector object named "d18Ow" in the ShellChron functions within the R environment. I will make sure to mention this specifically on revision.

- A sinusoidal temperature curve is not a valid assumption in many tropical sites where the temperature annual cycle is bimodal. The growth curve is also a strong a priori assumption that cannot be tested. Growth parameters often evolves through ontogeny. Uncertainties estimated by the model do not include the uncertainty related to these assumptions.

These are all fair comments. I fully acknowledge that the sinusoidal temperature curve and skewed-sinusoidal growth rate curves are approximations of reality. These approximations are discussed in detail in Judd et al. (2018), where they are first defended. In fact, the growth and temperature sinusoids are very flexible and can be modelled to a wide array of data shapes. Their use as target curves is therefore not a very restricting assumption. The fact that growth parameters evolve through ontogeny does not influence the accuracy of ShellChron, because each modelled window is independent. Growth parameters can (and will) change throughout the $\delta^{18}O$ record, and this change can be modelled by ShellChron. Indeed, there are situations in which the combination of sinusoids is not valid, and the tropical bimodal temperature seasonality is a good example. The reviewer therefore correctly highlights that ShellChron can only be used is archives where the sinusoidal $\delta^{18}O$ variability can be assumed to be annual. In reply to this comment, I will more clearly highlight this caveat of ShellChron in the revised manuscript.

- It is finally unclear whether ShellChron requires annual marker to be defined or not. While the text says it is not required (can be used with archives without annual markers), Figure 2 suggests otherwise, and it is not clear how the moving window size is determined if annual markers are not provided. If annual markers are provided, how is the uncertainty of this date dealt with?

This is a valid point, and I realize that this is not sufficiently clear from the text (see lines 233-241). In short, I tried to make ShellChron as independent from user-provided yearmarkers as possible. The yearmarkers in the input data are only used to constrain the minimum lengths the modelling windows require to contain a full year of growth. The sizes of the moving windows are determined at the beginning of the model routine (in the "data import" function) by linearly interpolating the distance between user-provided yearmarkers. After this step, the yearmarkers are not further used in ShellChron. This allows the model to freely assign the annual growth per window. This is one of the characteristics that distinguishes ShellChron from the Judd et al. (2018) model, which does use the yearmarkers to divide the $\delta^{18}O$ curve into annual chunks before running the model. In the revised version, I will make sure that the use of the yearmarkers is clearly explained.

- Figure 1 should be more explicit about input data and output data.

The figure already shows which data goes into the model and which comes out and illustrates the flow of pieces of data within and between functions in the model. To answer to the reviewer's comment, I will spell out the exported data types more specifically in the figure. I invite any suggestions on how to further specify details on input and output.

- Figure 2: "simulated parameters" should be defined more explicitely.

I will name the individual simulated parameters (e.g. "$\delta^{18}O$", "temperature", "age", "SD on age", etc.) directly in the figure and caption.

- Tests are performed with very high resolution records (>23 datapoints per year), while the resolution in seasonally resolved records is generally limited to 10-12 datapoints per year for cost optimization reasons. The model should be tested with lower resolution records, in accord with real paleoclimate practice.

This is an excellent suggestion, and I will include an example of a lower resolution dataset to provide a more realistic benchmark for testing ShellChron. Note that real examples were already included in the study (see **Fig. 4**), but I concur that the sampling resolution in these cases is probably higher than average.

- Case 1 is almost an ideal case. Surprisingly, the standard deviation of the chronology is more than a month. It seems that a simple linear interpolation would yield a better precision.

Case one represents an ideal case in which growth rate does not vary seasonally or change along the record. In this special case, linear interpolation indeed outperforms the sinusoidal model, because the growth rate is constant. Unfortunately, such cases are rare (if they exist at all) in nature, so the fact that the linear interpolation outperforms the results of ShellChron does not disprove that sinusoidal modelling of $\delta^{18}O$ is superior to linear interpolation in natural cases. Note that the standard deviation on the chronology in ShellChron includes uncertainty on the optimization routine as well as on the measurements of $\delta^{18}O$ and the position of the sample (in sampling domain). In light of these uncertainties, it seems that an uncertainty of more than a month is realistic.

- Testing with case 1 and Texel presented in Figure 3 shows that the model produced by ShellChron systematically underestimates the annual cycle amplitude (in contradiction with the text which says that there is no systematic seasonal bias). This is not necessarily a problem for the depth-age model, but this is not supposed to happen based on the calculation description and points to a potential issue in the code.

I fully agree with this comment, and Reviewer #1 highlighted a few issues with the model that may cause this underestimation (see reply to their comments above). In short, I will experiment with the parameters restricting the SCEUA optimization, the use of the sinusoidal regression and the implementation of weights in averaging the overlapping model solutions to test whether the choice of parameters in ShellChron introduces artifacts on individual model simulations (e.g. flat lines on modelled $\delta^{18}O$ and dampened seasonal amplitude) which propagate into the model result. The results of these tests will be added to the revised manuscript. In addition, I will add a step-by-step guide through the model functions using a more common, low resolution $\delta^{18}O$ dataset to illustrate the effect of the

various modelling steps. Finally, I will discuss the effect of changing or introducing model parameters on the age-depth result and check model residuals for systematic offsets in ShellChron's age results.

- The model is doing poorly with the Texel case. The author acknowledges that errors occur because of monthly scale variability, which is actually what happens in the real world. The chronology of the Texel case would be accurately obtained with seasonal anchor points and linear interpolation. I have not entered into the code but it seems that it needs to be improved.

ShellChron indeed simulated a jump in the age-depth relationship for the Texel sample. Under normal circumstances, such a jump of one full year in the model could be easily detected and removed from the data (as discussed in lines 355-361) or amended by re-running the model. However, I decided to use this result to illustrate the types of issues the user could run into while implementing a model like ShellChron without proper supervision of the result.

In all cases in this manuscript, it is assumed that no environmental data is available. In analyzing the Texel dataset without *a priori* knowledge about the environmental temperature and salinity changes, linear interpolation between the only known anchor points (the seasonal $\delta^{18}O$ extremes) would yield less accurate results than ShellChron. Not only is the Texel case marked by seasonal changes in both temperature and salinity (and therefore $\delta^{18}O_w$), but its growth rate also varies seasonally and decreases along the record. These nuances cannot be captured by linear interpolation between seasonal extremes, making sinusoidal models like ShellChron or the model in Judd et al. (2018) the better option.

- The author says in the text that the model can be used with speleothems, in contradiction with the testing which concludes that the performance is too low with the speleothem case.

The speleothem case is included as a worst-case example, and ShellChron's performance in this case is indeed low. However, in the discussion I highlight that age modelling in speleothems is possible in principle if the $\delta^{18}O$ record is good enough (see lines 181-184). I concur that these statements may come across as contradictory. Therefore, I will nuance my discussion of the application of ShellChron on speleothem records by more clearly stating the conditions these records should meet before results from ShellChron (or other sinusoidal $\delta^{18}O$ modelling) can be relied upon.

- In the test with real-world records, the performance of the model is evaluated using chronologies reconstructed using simple fit and interpolation method, which seems like an implicit acknowledgement that these simple techniques are superior.

Linear interpolation is only used to compare to the ShellChron results for the speleothem record (see **Fig. 4C**). The reason for this is that no sub-annual age model is available for this speleothem. By no means do I mean to suggest that speleothems grow through constant, year-round linear extension. In fact, drip water supply to (and therefore growth in) most speleothems varies seasonally (see Baldini et al., 2008; Van Rampelbergh et al., 2014; Vansteenberge et al., 2019; see lines 181-184 and lines 488-500). Therefore, sinusoidal models like ShellChron likely result in more realistic sub-annual growth rate reconstuctions than linear interpolation. I therefore think that the statement by the reviewer is not correct.

---

## Author Response (AR1)

Dear Geoscientific Model Development Editorial Board, Dear reviewers,

My apologies for the delay in revising my manuscript and replying to the review comments. Let me start by thanking both reviewers for their critical and constructive feedback on my manuscript. Both reviewers are quite critical on the ShellChron approach and raise valid concerns about its performance. The comments from the reviewers motivated me to revise not only the manuscript but also the R code of ShellChron, which resulted in several substantial improvements. I believe my updates to the model code and manuscript text provide a suitable reply to the issues raised by the reviewers and significantly improve the performance of ShellChron. Since these improvements required me to re-run all examples I use in my manuscript, the revision took longer than expected, hence my delay.

Below, I will first briefly summarize the improvements I made to ShellChron and how they influence the presentation of the model in my manuscript. Next, I attempt to summarize the main concerns raised by the reviewers and reply in general. I will then provide a point-by-point reply to the comments posed by the reviewers. Finally, I enclose within this resubmission a version of the manuscript in which track changes are enabled to show where changes were made to the manuscript text.

**Model revision**

In reply to the review comments, I have implemented options in ShellChron that enable the user to set the parameters of the SCEUA algorithm (see **section 4.1** in the revised manuscript text and Duan et al., 1992 for details) as well as to disable the sinusoidal regression ShellChron carries out before each optimization step to pick better initial values for the growth rate and temperature sinusoid parameters (see **section 3** and **equation 9** in the revised manuscript text). Changing these parameters allowed me to test and discuss their effect on the model result based on a short example dataset (**Test case**) to formulate recommendations for choosing the right parameters for modelling (see discussion in **section 4.1** of the revised manuscript).

Another major issue raised by both reviewers is that the previous version of ShellChron generally underestimated the seasonal $\delta^{18}O_c$ range. In addition, the previous version of the model returned sequences of constant $\delta^{18}O_c$ values in individual modelling windows, causing $\delta^{18}O_c$ estimates in these parts of the window to deviate significantly from the input $\delta^{18}O_c$ data. Close inspection of the model code revealed that these issues were related, and were caused by a combination of issues in the *run_model* and *cumulative_day* functions:

1. The boundaries on the growth rate sinusoid were set too liberally, causing growth rates to vary by several orders of magnitude within years and allowing growth rate variability over time to be nearly flat or vertical. The *run_model* function was modified to make the growth rate boundaries more restrictive, and dependent on the estimated initial value for the mean growth rate. The new boundaries still allow growth rate to vary by a factor 10 within a year, which should cover the natural variability of growth rates in $\delta^{18}O_c$ archives.
2. Constant $\delta^{18}O_c$ solutions in modelling windows were often a result of the number of years of $\delta^{18}O_c$ that were modelled within a window. If the window contained more time than expected the samples outside the $\delta^{18}O_c$ curve were assigned a constant value, resulting in a flat $\delta^{18}O_c$

profile. This issue also explained why the mean modeled $\delta^{18}O_c$ values underestimated the true $\delta^{18}O_c$ variability in the record.

3. Slight modifications were made to the *cumulative_day* function that aligns results from individual modelling windows to a common time axis. The previous version of ShellChron added too many or too little days to the age results in modelling windows in which either none or multiple year transitions occur. This results in sudden jumps in time in the age-distance relationship, such as the one in the **Texel** dataset discussed in the previous manuscript version (former **Fig. 3**, now updated to **Fig. 6** with re-run examples using the new model version, resolving the jump in time).

As a result of these improvements, the performance of ShellChron in terms of $\delta^{18}O_c$ approximation, age uncertainty, seasonal bias in age offset and even run time has significantly improved. To illustrate this, I include a figure showing the results of running ShellChron before and after the improvements on the new **Test case** dataset below.

[Figure]

Figure showing the difference in $\delta^{18}O_c$ approximation (top), age uncertainty (middle) and seasonal bias in age offset (bottom) between ShellChron runs on the same dataset (**Test case**) before (left) and after (right) the improvements to the model code summarized above.

**Reply to general review comments**

Firstly, I do not agree with Reviewer #2 that the model is oversophisticated and that it has no advantage over linear interpolation between seasonal extremes. As stated by Reviewer #1, there exist many seasonal archives with strong seasonal variability in extension rate (e.g. low-latitude bivalve shells). In most cases, approximating the seasonal curve through sinusoidal modelling, as is done by both ShellChron and the model by Judd et al. (2018), yields superior results over linear interpolation between seasonal extremes (see discussion in revised manuscript **section 5**).

That said, the reviewers highlight several key characteristics in ShellChron's output which indicate either a lack of clarity in the explanation of the calculations or sub-optimal implementation of the model. The reviewers give helpful suggestions on how the explanation can be improved, for example by including a step-by-step guide through a lower resolution example (**Test case**, see **section 3** and **Figure 4** in the revised manuscript) and clearer explanation of how the parameters of ShellChron (e.g. days per year and length of modelling windows) are defined. These issues were solved through an update of the manuscript text.

Issues related to the implementation of the model include the underestimation of the seasonal temperature amplitude highlighted by both reviewers, the occurrence of "flat lines" in individual modelling windows and the occurrence of high-frequency variability in the modelled temperature and growth curve after weighing and combining results from multiple modelling windows. Issues pertaining to the shape and fit of individual modelling windows relate to the way the SCEUA algorithm is implemented, and how its starting parameters are set in ShellChron (see comment by Reviewer #1). The overall match of the model to the multi-year $\delta^{18}O$ curve depends on the combination of consecutive growth years and the recognition of year transitions in the model. These aspects of ShellChron were all improved by updates to the code (see above). For the revised manuscript version, all examples are re-run using the updated model and the figures are updated with the new output.

**Point-by-point reply**

**Reviewer #1**

In *ShellChron 0.2.8: A new tool for constructing chronologies in accretionary carbonate archives from stable oxygen isotope profiles*, de Winter presents a model for temporally aligning sclerochronologic data. The approach expands on the growth rate model of Judd et al. (2018), with three notable improvements: (1) a sliding window approach that allows for more continuous age transformation of multi-year records, (2) a modular code design that increases usability and flexibility of application (e.g., user-specified proxy-to-temperature transfer function), and (3) propagation of error and uncertainty.

In premise, the approach offers an innovative solution to a complex problem and promises to elegantly addresses the key limitations of previous growth models. However, after reading the manuscript several times and going through the supplement, I am still left with several questions about the practical application of the approach, including but not limited to:

(1) **Attenuated amplitude of the modelled isotope data**: Why do the modelled curves seem to consistently underestimate the known isotopic range (e.g., **Fig 3**)? The manuscript explains that each

window is designed to include at least one full year of growth (L238), meaning that at least one (if not both) seasonal extremes should be included in any given window. Additionally, the stated benefit of fitting a (linearized) sinusoid to the data in the window (in the depth domain) prior to iterating the temperature and growth rate functions is to set realistic starting parameters and bounds (L267-272). Given this, it seems the modelled isotope curve of each window should accurately capture the seasonal range of values. This point is illustrated in **Fig. S1**, which shows the known and weighted mean modelled isotope curves for the Texel high resolution scenario (**Fig. S1A**) and the residuals (**Fig. S1B**). Consistent with **Fig. 3**, the residuals are normally distributed and center at 0‰; however, they exhibit a non-random distribution, paralleling the seasonal signal – a direct result of systematically underestimating seasonal extremes. This is important as it not only detracts from the potential climatological and biological insights from the temperature and growth rate functions, it also leads to spurious intra-annual temporal alignment of the data.

This is a valid point. Indeed, the previous version of ShellChron systematically underestimated the total seasonal amplitude in the data. Part of the reason for this lies in the (weighed) averaging of overlapping model windows. As illustrated in **Fig. 3**, individual model windows are not very sensitive to single datapoints, causing a slight smoothing of the seasonal $\delta^{18}$O curve. This effect is exacerbated by the weighing of the $\delta^{18}$O data within model windows, which causes datapoints on the edge of the model window to have lesser influence on the shape of the modelled curve. This weighing is implemented to prevent outliers in the data to influence the model outcome. The downside of this design choice is that individual datapoints in the extreme seasons (especially clear in the Texel case) fail to draw the simulation to the recorded maxima and minima in $\delta^{18}$O. A re-run of the examples with the updated model code (see above) shows that the issues resulting from averaging values from consecutive windows can be overcome by improving individual model window results through changes in the *run_model* and *cumulative_day* functions. These updates eliminate the underestimation of seasonal $\delta^{18}$O$_c$ extremes (see **Fig. 6** and **Fig. 7** in the revised manuscript).

Another factor that may contribute to the poor fit quality of some model windows might be the freedom given to the SCEUA algorithm. Currently, the algorithm is set to stop optimization when function values ($\delta^{18}$O) and model parameters do not change by more than 0.01% between consecutive model runs ("pcento" and "peps" parameters in the *run_model* function), and the model is limited to 10000 function evaluations ("maxn" in "run_model.r") to limit processing time. Since large $\delta^{18}$O datasets and overlapping modelling windows cause full ShellChron runs to already take over one hour on a modern personal computer in the current setup (as noted by Reviewer #2), limiting processing time is valuable. The sinusoidal fit was implemented to help the algorithm to find better starting values and cause the SCEUA optimization to converge within the set parameters. In the updated version of ShellChron, I implemented an option in ShellChron for the user to set the constraints on the SCEUA optimization algorithm manually and to switch the sinusoidal fit off. To evaluate if the limitations placed on the SCEUA algorithm and/or the use of sinusoidal fitting cause underestimation of the seasonal $\delta^{18}$O amplitude, a range of SCEUA settings were tested with and without sinusoidal fit on the small **Test case** dataset (see **section 4.1** in the revised manuscript).

While the underestimation of the seasonal amplitude in $\delta^{18}$O data does indeed dampen the seasonal cycle of the modelled temperature curve, it should not negatively affect the modelled age-depth relationship (as was recognized by Reviewer #2, see below). To demonstrate the improvement in the estimation of seasonal $\delta^{18}$O$_c$ extremes and discuss any systematic (seasonal) offsets in the age alignment of the $\delta^{18}$O data, I added a seasonal breakdown of the age offset to **Fig. 6** (previously **Fig. 3**, containing the **Case 1** and **Texel** results). Note that similar seasonal age offset plots for the other cases are also provided in **SI12**.

**(2) Number of growing days**: In looking through the supplement for the Texel high resolution scenario, more than 35% of the windows predict that the duration of growth captured by the isotope data in each window is less than 7 days (as per '*Day_of_year_raw.csv*'). Again, if each window spans at least one year and realistic bounds have been set for the temperature and growth rate functions, then why is growth condensed into such a short duration?

The reviewer refers to the "flat lines" in some individual modelling windows, where ShellChron fails to properly approximate the seasonal $\delta^{18}O$ curve (see following comment). This issue is related to the previous point raised by the reviewer, and results from several issues in the model code which have been resolved in the new version of ShellChron (see above). If not properly recognized and removed, these flat model results in the previous model version resulted dampening of the seasonal $\delta^{18}O$ curve by the model, because they are included in the average of the modelled $\delta^{18}O$ curve.

**(3) Shape of the modelled isotope curves in the depth and time domains**: Why do some modelled isotope curves exhibit high frequency variability (e.g., Window 3 of the Texel high resolution scenario; **Fig. S2**)? My understanding is that the modelled curves derive from the sinusoidal temperature function and the skewed sinusoidal growth rate function, both of which exhibit monotonic increases or decreases between their peaks and troughs. Are these wiggles a function of the weighting? If so, why is it not consistent across all windows? Also, as alluded to above, many of the windowed curves exhibit long flat lines at their start and/or end, indicating that the temperature and growth rate function bounds might be impeding a better fit. I suspect there might be an error in the Case 1 and Texel examples, as some of the concerns addressed above don't appear in the Coral, Oyster, and/or Speleothem examples. (Could the upper amplitude bound accidentally been set at $T_{amp}/2$ in these examples?) However, it is difficult to (a) ensure that I fully understand the methodology and (b) evaluate the fidelity and applicability of the approach while these questions linger.

Individual model simulation ("windows") are constrained by the two sinusoids and will therefore not exhibit high-frequency variability. On checking the supplementary file **SI7** from which the data in the reviewer's **Fig. S2** is obtained I discovered that the high-frequency variability only occurs in modelled $\delta^{18}O$ data from Window 3, and only in the high-resolution Texel dataset. None of the modelling window results from any of the tested datasets (see former **SI7** and **SI8**, and **SI7-10** of the revised supplement) exhibit this variability. The incompatible data in Window 3 of the high-resolution test on the Texel data resulted from a copying error. My apologies for the inconvenience. I provided a clean export of the raw results of the updated model on all test datasets (see **SI7-10**).

**Other general comments**

• The manuscript would benefit from a detailed guided example. It is difficult to clearly and concisely convey complex mathematical model. Rather than presenting the Case 1 and Texel examples (which ultimately don't add much insight that can't be gleaned from the real-world examples), I would suggest creating a shorter (<5 yearlong) virtual dataset and stepping the reader through 2 or 3 window examples, showing the temperature and growth rate functions for each window and comparing those to the known values.

This is a great suggestion, and I have now added the smaller **Test case** dataset in the revised version of the manuscript to provide a step-by-step guided example through the model using a realistic $\delta^{18}O$ record (see **Fig. 2** and **Fig. 4** in the revised manuscript).

• It would be useful to address how the moving window approach is impacted by changes to the number of samples per year change and/or interannual extension rate. In practice, it can sometimes be difficult to sample at a constant rate and/or samples are sometimes lost or too small to run. How would the moving window approach adapt to an example where the first year had 15 samples and the second had only 8? Similarly, how are the results impacted if there is a large change to the interannual extension rate (as is observed in some low latitude bivalves)?

A good suggestion, and I implemented it by testing ShellChron on the **Test case** dataset, which has a lower and variable sampling resolution and growth rate.

• Can you please clarify whether the period of the window (in the time domain) is defined during the linearized sinusoidal fit or if it is fixed at 365 days?

The period of one year, as defined by the model outcome, is set at 365 days (although it is possible for the user to override this default through the "T_per" parameter, see lines 309-310 in the revised manuscript). Window length is defined based on linear interpolation between the user provided YEARMARKERS such that each window contains at least one growth year. The length of a full window can therefore be longer than 365 days, as the length of one year in sampling direction is determined by the model outcome based on the $\delta^{18}O_c$ data. An initial value for the period in sampling direction is found through the sinusoidal fit before the SCEUA algorithm optimizes the growth rate and temperature sinusoids. These points are clarified in the revised manuscript (**section 3**).

**Line-by-line comments**

L22: This notation is not defined until L70; I'd suggest rephrasing to 'oxygen isotope records'

I will rephrase this accordingly in the Abstract.

L39: Change 'time' to 'temporal'

I will rephase to "temporal"

L54: Delete 'of'

"of" will be deleted

L54: Change 'greenhouse warming' to 'anthropogenic warming' or something of the like – we're still a far way off from a true ice-free greenhouse climate

Agreed, I did not want to suggest that we are in a greenhouse state, rather refer to "warming by greenhouse gases". I agree that "anthropogenic (global) warming" is better and will rephrase accordingly.

L58: Change 'by' to 'at'

"by" will be rephrased to "at"

L71: Change 'by' to 'at'

"by" will be rephrased to "at"

L73: 'one being dominant over the other' – I'm not sure I completely agree; it can be quite difficult to disentangle these signals with, for example, low latitude, shallow bivalves and corals that are subject to seasonal precipitation; perhaps revise to 'with one generally being dominant over the other'

Agreed, I will rephrase to "with one generally being dominant over the other".

L91: Change 'causes them to rely on' to 'require'

I will rephrase to "requires"

L109: Define SCE-UA (Shuffled Complex Evolution model developed at the University of Arizona)

The abbreviation will be defined here.

L112: Sample trajectories are referred to as "depths" in the text of the manuscript but "lengths" in the figures (I'd suggest opting for "sample distance", as that term is more widely applicable ("depth" is an odd term in reference to bivalves)

This is a good point, and I will refer to "sample distance" or "distance in sampling direction" throughout the manuscript.

L138: Rephrase 'the question which'

I will rephrase to "Which carbonates are…"

L138 – 154: The original foraminifera references should be Spero et al. (1997) and Zeebe (1999). However, I'd contend that these studies do not suggest that foraminifera are precipitated out of equilibrium with their environment, but rather that additional environmental variables pertaining to the carbonate system (e.g., pH) contribute to oxygen isotope value of carbonate. This discussion is important (!) but highly nuanced, and I'm not sure this is the appropriate platform to address it. I'd suggest removing this section or moving it to the supplement; the important part of this paragraph as it pertains to the work presented in the paper is that ShellChron permits the user to define their desired transfer function.

I agree that this is not the place to discuss the equilibrium fractionation of foraminifera or the effects of environmental parameters on their $\delta^{18}O$ value. Therefore, this paragraph was shortened, focusing on the importance of ShellChron's flexibility in choosing different transfer functions for the $\delta^{18}O$-temperature relationship. The background paragraph on equilibrium fractionation was moved to the **Supplementary Discussion**.

L162-163: Rephrase 'temperature evolution'

This will be rephrased to "the evolution of the calcification temperature"

L168: Change 'priory' to 'priori'

This will be rephrased accordingly.

L206 (and all other instances): I don't know that 'depth' is the correct word choice; bivalves aren't generallydiscussed in terms of a depth domain, but instead a sampling 'distance' along a growth axis

See above, I agree and will refer to "distance (in sampling direction)" throughout the manuscript.

L211,212: Change both instances of 'were' to 'are' for consistency of tense throughout the paragraph

This will be rephrased for consistency.

L316: delete 'in the'

This will be rephrased to "Wadden Sea (North Netherlands; Texel; see details in…

**Reviewer #2**

The paper presents the details of ShellChron, an R package that calculates the inner chronology of seasonally resolved oxygen isotopic records such as those obtained from corals, mollusks or some speleothems. ShellChron does so by calculating a d18O curve combining an assumed sinusoidal temperature curve, a quasi-sinusoidal growth curve, and a paleo-temperature equation, fit onto the d18O record in moving windows. The performance of the model is evaluated using synthetic and real records.

While I acknowledge the importance of the inner chronology issue in the study of seasonal scale processes from paleoclimate records, ShellChron seems like a heavy oversophisticated solution (several hours of calculations per depth-age model) with no clear advantage compared to simple techniques like matching the d18O record with local temperature curve or a simple linear interpolation between seasonal isotopic extrema used as anchor points.

I agree that, in modern cases where local temperature and salinity curves are available, a modelling approach like ShellChron or the model in Judd et al. (2018) is not necessary, because the $\delta^{18}O$ data can be aligned to those known temperature and salinity curves. However, in cases where "true" environmental data are not available, modelling the seasonality curve has clear advantages over simple linear interpolation between seasonal extremes. As discussed in lines 84-99 of the revised manuscript, in most shallow marine carbonate archives both temperature and growth rate describe quasi-sinusoidal patterns. Linear interpolation between maxima and minima (assuming constant growth rates in between) cannot approximate the growth rate in these records as well as a sinusoidal curve (see discussion in Judd et al. 2018). Therefore, in absence of "true" temperature and salinity records (i.e. in fossil archives), the sinusoidal modelling approach is a much better assumption for age-modelling than linear interpolation.

The author interprets too positively the test results. In my view, the tests indicate low performance in terms of dating precision despite the high quality of the records (synthetic or real), which questions the usefulness of the whole package.

In my discussion in relation to former **Figures 3-5** (now **Figs. 6-8**), I honestly discuss the uncertainties of ShellChron in terms of reproducibility of the model result (precision) as well as agreement with the "true" age-distance relationship of the archive (accuracy). The ages found by the updated ShellChron model tend to have uncertainties of below 30 days, which is sufficient for constraining the seasonal cycle (see discussion in **section 5**, lines 565-579). The advantage of ShellChron is that this uncertainty is known and quantifiable. In previous models, and especially when assuming constant growth between seasonal extremes (see above), this uncertainty is unknown. Furthermore, uncertainties calculated by ShellChron uniquely consider uncertainties in both the $\delta^{18}O_c$ measurement and the sample position, making these uncertainties seem higher than those in other models. If the reviewer believes the revised discussion of ShellChron's uncertainty is still insufficient, I invite any comments indicating where the results of ShellChron are interpreted too positively and how these improvements could be made.

- It is said in the text that a seasonally varying water d18O curve can be included but it is not clear how to do so in practice.

This is a valid point. In fact, the $\delta^{18}O_c$ curve can be provided as a vector object named "d18Ow" in the ShellChron functions within the R environment. This is now mentioned specifically in the revised manuscript (see lines 159-161).

- A sinusoidal temperature curve is not a valid assumption in many tropical sites where the temperature annual cycle is bimodal. The growth curve is also a strong a priori assumption that cannot be tested. Growth parameters often evolves through ontogeny. Uncertainties estimated by the model do not include the uncertainty related to these assumptions.

These are all fair comments. I fully acknowledge that the sinusoidal temperature curve and skewed-sinusoidal growth rate curves are approximations of reality. These approximations are discussed in detail in Judd et al. (2018), where they are first defended. In fact, the growth and temperature sinusoids are very flexible and can be modelled to a wide array of data shapes (see example in **Fig. 4**). Their use as target curves is therefore not a very restricting assumption. The fact that growth parameters evolve through ontogeny does not influence the accuracy of ShellChron, because each modelled window is independent. Growth parameters can (and will) change throughout the $\delta^{18}O$ record, and this change can be modelled by ShellChron. Indeed, there are situations in which the combination of sinusoids is not valid, and the tropical bimodal temperature seasonality is a good example. The reviewer therefore correctly highlights that ShellChron can only be used is archives where the sinusoidal $\delta^{18}O$ variability can be assumed to be annual. In reply to this comment, I highlighted this caveat of ShellChron more clearly in the revised manuscript (lines 623-628 of the revised manuscript).

- It is finally unclear whether ShellChron requires annual marker to be defined or not. While the text says it is not required (can be used with archives without annual markers), Figure 2 suggests otherwise, and it is not clear how the moving window size is determined if annual markers are not provided. If annual markers are provided, how is the uncertainty of this date dealt with?

This is a valid point, and I realize that this was not sufficiently clear from the text. In short, I tried to make ShellChron as independent from user-provided yearmarkers as possible. The yearmarkers in the input data are only used to constrain the minimum lengths the modelling windows require to contain a full year of growth. The sizes of the moving windows are determined at the beginning of the model routine (in the "data import" function) by linearly interpolating the distance between user-provided yearmarkers. After this step, the yearmarkers are not further used in ShellChron. This allows the model to freely assign the annual growth per window. This is one of the characteristics that distinguishes ShellChron from the Judd et al. (2018) model, which does use the yearmarkers to divide the $\delta^{18}O$ curve into annual chunks before running the model. In the revised version, I more clearly explained the use of the yearmarkers (see lines 239-249, 257-258, 536-538 and **Fig. 2** in the revised manuscript).

- Figure 1 should be more explicit about input data and output data.

The figure already shows which data goes into the model and which comes out and illustrates the flow of pieces of data within and between functions in the model. To answer to the reviewer's comment, I spelled out the exported data types more specifically in the figure (see revised **Fig. 1**). I invite any suggestions on how to further specify details on input and output.

- Figure 2: "simulated parameters" should be defined more explicitly.

I named the individual simulated parameters (e.g. "δ$^{18}$O", "temperature", "age", "SD on age", etc.) directly in **Fig. 3** in the revised manuscript.

- Tests are performed with very high resolution records (>23 datapoints per year), while the resolution in seasonally resolved records is generally limited to 10-12 datapoints per year for cost optimization reasons. The model should be tested with lower resolution records, in accord with real paleoclimate practice.

This is an excellent suggestion, and I included an example of a lower resolution dataset (**Test case**) to provide a more realistic benchmark for testing ShellChron (see **Figs. 2** and **4**). Note that real examples were already included in the study (see **Fig. 7**), but I concur that the sampling resolution in these cases is probably higher than average.

- Case 1 is almost an ideal case. Surprisingly, the standard deviation of the chronology is more than a month. It seems that a simple linear interpolation would yield a better precision.

Case one represents an ideal case in which growth rate does not vary seasonally or change along the record. In this special case, linear interpolation indeed outperforms the sinusoidal model, because the growth rate is constant. Unfortunately, such cases are rare (if they exist at all) in nature, so the fact that the linear interpolation outperforms the results of ShellChron does not disprove that sinusoidal modelling of δ$^{18}$O is superior to linear interpolation in natural cases. Note that the standard deviation on the chronology in ShellChron includes uncertainty on the optimization routine as well as on the measurements of δ$^{18}$O and the position of the sample (in sampling domain), and that this uncertainty was significantly reduced in the new model version. In light of these uncertainties on the input variables, it seems that ShellChron's uncertainty of ~12 days is realistic. In addition, I provide a comparison between the real age offset and the uncertainty given by the model in **Fig. 8C**.

- Testing with case 1 and Texel presented in Figure 3 shows that the model produced by ShellChron systematically underestimates the annual cycle amplitude (in contradiction with the text which says that there is no systematic seasonal bias). This is not necessarily a problem for the depth-age model, but this is not supposed to happen based on the calculation description and points to a potential issue in the code.

I fully agree with this comment, and Reviewer #1 highlighted a few issues with the previous version of the model that may cause this underestimation (see reply to their comments above). Several improvements to ShellChron now resolve these issues and result in negligible underestimation of the δ$^{18}$O$_c$ amplitude and no seasonal bias on the model (see **Model improvements** paragraph above and answers to comments raised by Reviewer #1). In addition, I added a step-by-step guide through the model functions using the **Test case** dataset to illustrate the effect of the various modelling steps (see revised manuscript **Figs. 2** and **4**). Finally, I discussed the effect of changing or introducing model parameters on the age-depth result and check model residuals for systematic offsets in ShellChron's age results (see **Fig. 5** in the revised manuscript and discussion in **section 4.1**).

- The model is doing poorly with the Texel case. The author acknowledges that errors occur because of monthly scale variability, which is actually what happens in the real world. The chronology of the Texel case would be accurately obtained with seasonal anchor points and linear interpolation. I have not entered into the code but it seems that it needs to be improved.

The previous version of ShellChron indeed simulated a jump in the age-depth relationship for the **Texel** sample. This error resulted from a combination of issues in the *cumulative_day* function which have been resolved in the updated version of ShellChron (see **Model improvements** paragraph and **Figs. 6-7** in the revised manuscript). In all cases in this manuscript, it is assumed that no environmental data is available. In analyzing the **Texel** dataset without *a priori* knowledge about the environmental temperature and salinity changes, linear interpolation between the only known anchor points (the seasonal $\delta^{18}O$ extremes) would yield less accurate results than ShellChron. Not only is the **Texel** case marked by seasonal changes in both temperature and salinity (and therefore $\delta^{18}O_w$), but its growth rate also varies seasonally and decreases along the record. These nuances cannot be captured by linear interpolation between seasonal extremes, making sinusoidal models like ShellChron or the model in Judd et al. (2018) the better option.

- The author says in the text that the model can be used with speleothems, in contradiction with the testing which concludes that the performance is too low with the speleothem case.

The **speleothem** case is included as a worst-case example, and ShellChron's performance in this case is indeed low. However, in the discussion I highlight that age modelling in speleothems is possible in principle if the $\delta^{18}O$ record is good enough (see lines 589-606 in the revised manuscript text). I concur that these statements may come across as contradictory. Therefore, I nuanced my discussion of the application of ShellChron on speleothem records by more clearly stating the conditions these records should meet before results from ShellChron (or other sinusoidal $\delta^{18}O$ modelling) can be relied upon (see revised **section 5**).

- In the test with real-world records, the performance of the model is evaluated using chronologies reconstructed using simple fit and interpolation method, which seems like an implicit acknowledgement that these simple techniques are superior.

Linear interpolation is only used to compare to the ShellChron results for the speleothem record (see **Fig. 7C** in the revised manuscript). The reason for this is that no sub-annual age model is available for this speleothem. By no means do I mean to suggest that speleothems grow through constant, year-round linear extension. In fact, drip water supply to (and therefore growth in) most speleothems varies seasonally (see Baldini et al., 2008; Van Rampelbergh et al., 2014; Vansteenberge et al., 2019; see discussion lines 589-606). Therefore, sinusoidal models like ShellChron likely result in more realistic sub-annual growth rate reconstuctions than linear interpolation. I therefore think that the statement by the reviewer is not correct.

---

## Referee Report (RR1)

**Rereview of *ShellChron 0.4.0: A new tool for constructing chronologies in accretionary carbonate archives from stable oxygen isotope profiles* by de Winter**

Overall, I am largely satisfied with the revisions made by de Winter to *ShellChron 0.4.0: A new tool for constructing chronologies in accretionary carbonate archives from stable oxygen isotope profiles*. Below I highlight four general comments, as well as some specific editorial comments. Pending minor-to-moderate revisions, I believe this manuscript will be suitable for publication in *Geoscientific Model Development*.

**General comments**

- **Fidelity of the examples**: My original main concern regarding the attenuation of the seasonal amplitude in the Case 1 and TEXEL examples seems to be resolved in this revision. However, the *Crassostrea gigas* example still shows some odd solutions (attenuation in Year 1 and multiple vertical lines in Year 2 in Figure 7B) not present in any of the other examples, that should be addressed. I'm also still slightly concerned by the systematically season residuals in Case 1 (Figure 6C; which also don't seem to fully match the distribution in Figure 6B). This seems to imply that something (e.g., bounds on the sine parameters, weighting of the windows) may be inducing a nonrandom bias. I encourage the author to explore this a bit more to determine where and why such biases occur, so that future users of the model have a better sense of whether such biases might affect their data.

- **Test 1 Example:** I'm a bit confused by the example – if each window is supposed to span at least one year (L243-245) and the input growth rate of Test 1 was sinusoidal with no growth cessation (Figure 2A), why do the x-axes in Figure 4A,B only span ~250 days? This seems to imply that the model is imposing a growth cessation and condensing growth into 250 days. Is this the case? If so, can you achieve a better fit by changing some of the starting parameters of the model? Please address this is the text. It would also be helpful to also the "known" (i.e., input) temperature and growth rate curves on these panels for comparison.

- **The impact of cumulative offsets**: Can you speculate on why the date offsets in the TEXEL example are not centered at zero? Is this phase shift an artifact of the windowed approach? e.g., If you were to omit the data from Years 1-3 and instead just model Years 4-9, would the results be different and, if so, by how much?

    What I'm getting at here, is that while I fully appreciate the advantages of the window approach and the formation of a single continuous time series, it ends up largely ignoring the only *a priori* age information we have about age (i.e., the placement of year markers or identifiable peaks and troughs in the data, which ultimately provide an annual chronometer). Many seasonal paleoclimate studies now stack temporally resolved data onto a single seasonal cycle, rather than analyzing them in time series, to get a climatological average (e.g., Tierney et al., 2020; Judd et al., 2019). In this sense, the *a priori* identified year markers help to minimize the propagation of uncertainty across the age model (i.e., one spurious year doesn't result in compounding error). A major advantage of the approach presented here is that it generates a continuous record, but I'm interested in understanding if and how this may propagate error across the record. I'm not sure that there's an easy solution here, but I think that the sensitively of the results to cumulative offsets and the range

over which the data is modelled (e.g., full dataset vs. only a few select years) should be addressed in the discussion.

Lastly, can you explain how the offset from actual age was computed for the Judd et al. model? Because each year is run individually using this model, the year markers define the starting and ending point of discrete years – so theoretically offsets should never be greater than >1 year (as suggested in Figure 8B), provided that year markers are correctly defined. In this sense and building on what is discussed above, it would perhaps be useful to see the year-to-year offsets rather than (or in addition to) the cumulative offset.

- **Speleothem example**: In my opinion, the speleothem example is neither necessary nor constructive. As the model is currently written, it is not designed for speleothems (L167-170) and the speleothem example only serves to highlight this. In most cases, speleothems violate the assumption of sinusoidal oscillation in $\delta^{18}O$ and thus would likely benefit from a different age modelling approach. The manuscript is already quite long with several examples, and there's no need to add length by explaining why such systems are difficult to model (e.g., L477-496; L588-605).

**Specific comments**

L217: add comma between "included" and "which"

L358: change "all." to "al."

L444-446: awkward phrasing; consider revising to "The lower sampling resolution later in the record mutes this variability and further illustrates that..."

L468: "real" feels colloquial and nonscientific; I'd recommend changing "real" to "known", both in the text and the figures (e.g., "the "known" age of the samples in these natural carbonates is not truly known.")

L476: remove "very" (such descriptors are unquantifiable)

L671: change "eliminate" to "minimize"

Figure 2: change the blue bar to reflect the window used in the Figure 4 example.

Figure 7: for consistency with Figure 6, I suggest keeping the isotope residuals histogram in the upper left corner and the date residuals histogram in the lower right corner

**New references cited**

Judd, Emily J., Linda C. Ivany, Robert M. DeConto, Anna Ruth W. Halberstadt, Nicole M. Miklus, Christopher K. Junium, and Benjamin T. Uveges (2019) "Seasonally resolved proxy data from the Antarctic Peninsula support a heterogeneous middle Eocene Southern Ocean." *Paleoceanography and Paleoclimatology* 34(5).

Tierney, Jessica E., Christopher J. Poulsen, Isabel P. Montañez, Tripti Bhattacharya, Ran Feng, Heather L. Ford, Bärbel Hönisch et al. (2020). "Past climates inform our future." *Science* 370(6517).

---

## Author Response (AR2)

Dear Geoscientific Model Development Editorial Board, Dear reviewers,

I would like to thank both reviewers for their comments and the editor dr. Olivier Marti for his moderation of the review process of my manuscript. I am delighted to read that both reviewers react positively to the revisions made to my manuscript and model code over the last review round, and that the manuscript is now ready for publication after minor revisions. Below, I offer a point-by-point reply to the questions and comments raised by the reviewers. In addition, I include a revised version of my manuscript in which I have tracked my changes. In addition to the suggested changes, I now refer to the Judd et al. (2018) model as the "GRATAISS" model throughout the manuscript text ("Growth Rate and Temporal Alignment of Isotopic Serial Samples"; following Ivany and Judd, 2022) in addition to other minor textual changes which I encountered on re-reading the manuscript.

**Reviewer #1**
**Rereview of *ShellChron 0.4.0: A new tool for constructing chronologies in accretionary carbonate***
***archives from stable oxygen isotope profiles* by de Winter**

*Overall, I am largely satisfied with the revisions made by de Winter to ShellChron 0.4.0: A new tool for constructing chronologies in accretionary carbonate archives from stable oxygen isotope profiles. Below I highlight four general comments, as well as some specific editorial comments. Pending minor-to moderate revisions, I believe this manuscript will be suitable for publication in Geoscientific Model Development.*

I would like to thank Referee #1 for thoroughly and fairly re-reviewing my manuscript and model, and especially for their patience during the lengthy review process. As mentioned in reply to the previous round of comments, I am greatly indebted to the reviewer for their constructive criticisms on the first manuscript version which helped me improve both the manuscript and model to their current state. I am glad that my revision is positively received and grateful for the additional comments that helped me to finetune the manuscript for publication.

**General comments**
• **Fidelity of the examples**: My original main concern regarding the attenuation of the seasonal amplitude in the Case 1 and TEXEL examples seems to be resolved in this revision. However, the *Crassostrea gigas* example still shows some odd solutions (attenuation in Year 1 and multiple vertical lines in Year 2 in Figure 7B) not present in any of the other examples, that should be addressed.

Model accuracy is lower at the beginning and end of records, because the result in these places is based on a lower number of overlapping model windows. The accuracy also diminishes at the beginning and end of each modelling window (producing the "offshoots" in **Fig. 7B**), which is resolved by weighting model results at beginning and end of model less heavily. The lower sampling resolution of the oyster in first year and uncertainty and MC approach together make this start of the record more sensitive to random model uncertainty attenuating the record. I added some discussion of these results to section 4.3.

*I'm also still slightly concerned by the systematically season residuals in Case 1 (Figure 6C; which also don't seem to fully match the distribution in Figure 6B). This seems to imply that something (e.g., bounds on the sine parameters, weighting of the windows) may be inducing a nonrandom bias. I encourage the author to explore this a bit more to determine where and why such biases occur, so that future users of the model have a better sense of whether such biases might affect their data.*

Fig 6A-B shows that the bias occurs at peaks and troughs where small uncertainties in the position of samples drives larger uncertainties in age model. This effect is also visible in the $\delta^{18}O_c$ model fits (**Fig. 6A**). From **Fig. 6B** it shows that a few of these peak and valley datapoints drive the uncertainty in the monthly distribution of age inaccuracy, which is not normally distributed (see histogram in **Fig. 6B**). The $\Delta t$ distribution in **Fig. 6B** matches the distribution in **Fig. 6C**, clearly showing the occurrence of small outliers on the positive $\Delta t$ scale matching the minima in the $\delta^{18}O_c$ curve (and also the maxima in some places). Note that the offset is barely significant and the mean dt is not more than ±10 days. Nevertheless, these issues are now briefly discussed in section 4.2.

• **Test 1 Example:** I'm a bit confused by the example – if each window is supposed to span at least one year (L243-245) and the input growth rate of Test 1 was sinusoidal with no growth cessation (Figure 2A), why do the x-axes in Figure 4A,B only span ~250 days? This seems to imply that the model is imposing a growth cessation and condensing growth into 250 days. Is this the case? If so, can you achieve a better fit by changing some of the starting parameters of the model? Please address this is the text. It would also be helpful to also the "known" (i.e., input) temperature and growth rate curves on these panels for comparison.

The model does not impose growth cessation (see **Fig. 6B**), but coarse sampling means that the growth rate sinusoid is not accurately resolved everywhere within the window. Also, the difference between the first (point 1) and last datapoint (point 11) is not 365 because the next datapoint (point 12), which is precisely 365 days from the first point in the window, belongs to the new year and is hence not included in the window. Note that the modelled temperature curve closely approximates the known temperature. Longer modelling times should result in better approximation of the growth rate sinusoid (increasing maxn, see lines 301-303 and Fig. 5), however uncertainty in single windows is smoothed out by the moving window approach and weighting of solutions at the edge of windows, so the solution of one window not an fair measure of model performance. I added some discussion of this example clarifying these results at the beginning of section 4.2.

• **The impact of cumulative offsets**: Can you speculate on why the date offsets in the TEXEL example are not centered at zero? Is this phase shift an artifact of the windowed approach? e.g., If you were to omit the data from Years 1-3 and instead just model Years 4-9, would the results be different and, if so, by how much? What I'm getting at here, is that while I fully appreciate the advantages of the window approach and the formation of a single continuous time series, it ends up largely ignoring the only *a priori* age information we have about age (i.e., the placement of year markers or identifiable peaks and troughs in the data, which ultimately provide an annual chronometer). Many seasonal paleoclimate studies now stack temporally resolved data onto a single seasonal cycle, rather than analyzing them in time series, to get a climatological average (e.g., Tierney et al., 2020; Judd et al., 2019). In this sense, the *a priori* identified year markers help to minimize the propagation of uncertainty across the age model (i.e., one spurious year doesn't result in compounding error). A major advantage of the approach presented here is that it generates a continuous record, but I'm interested in understanding if and how this may propagate error across the record. I'm not sure that there's an easy solution here, but I think that the sensitively of the results to cumulative offsets and the range over which the data is modelled (e.g., full dataset vs. only a few select years) should be addressed in the discussion.

There is no cumulative offset in the model. The offset in the Texel example does not increase over record (accumulating uncertainty, see also histogram in **Fig. 6E**). Each window is fully independent from the previous window. Relative age results of windows are only later combined into continuous age-depth model by adding whole numbers of years, so there is no way the result of previous windows or years affects the seasonal phase of the next window.
The offset in the Texel example results from the way the real and modelled age-depth records are aligned at the same reference point in time. The small negative offset of the first datapoint from zero was thereby projected on the entire record (equally, without seasonal bias). I decided against scaling the entire modelled age downward to reduce this offset to prevent circular reasoning, but I will discuss this effect in the revised manuscript. Note that this choice of defining the "zero" of the age model result does not play a role in records with unknown ages, as the age result can be anchored to growth cessations or extreme values in $\delta^{18}O_c$. If all age windows were to be treated individually (as in the GRATAISS model), this "age alignment" decision will need to be made multiple times, potentially introducing uncertainty on year-by-year comparisons.
Finally, I agree that there are lots of uses for stacking data along the seasonal cycle (which does not require a continuous age record) as well as for continuous multi-year records. The advantage of ShellChron is that it allows data to be stacked relative to the seasonal cycle as well as providing a continuous record spanning multiple years. I now highlight this point in section 5.

Lastly, can you explain how the offset from actual age was computed for the Judd et al. model? Because each year is run individually using this model, the year markers define the starting and ending point of discrete years – so theoretically offsets should never be greater than >1 year (as suggested in Figure 8B), provided that year markers are correctly defined. In this sense and building

on what is discussed above, it would perhaps be useful to see the year-to-year offsets rather than (or in addition to) the cumulative offset.

To obtain cumulative age estimates from the Judd et al. model, which indeed treats years individually, I exported the age data from the model and detected transitions from one year over to the next. Individual years were linked by having the results of the next year "window" in the sequence starting in the next year (e.g. 365 days or more after the first datapoint in the current year), thereby retaining the seasonal phase of the datapoints and preventing age inversions. This procedure results in gaps of time being introduced in the age model result due to the results of two consecutive years not matching up perfectly, which accumulate along the record to result in offsets larger than 1 year (see **Fig. 8B**). I concur that these gaps are not really a result of the Judd et al. model but are inevitable when using it to create multi-year age models from $\delta^{18}O_c$ records. This issue is one of the main reasons why I set out to make ShellChron, as most $\delta^{18}O_c$ records record multiple years and applying the Judd et al. model often results in overlapping ages for consecutive $\delta^{18}O_c$ measurements (age inversions) or large time gaps between model years. Note that this issue may not be visible when stacking multiple years along a seasonal cycle, as is often done in paleoclimate studies, but this data treatment approach does not solve the issue: Age inversions, which result from inaccuracies in age modelling, may still be present at the beginning and end of the years, but a stacking approach will make it harder to detect them. The advantage of ShellChron is that the uncertainties at the edges of year windows are quantified and included in the model result, thereby making it possible to estimate the confidence of the age model and the likelihood that age inversions occur in the record from overlapping errors on results for consecutive datapoints.
This comment by the reviewer highlights an important difference between the two models which I agree needs some discussion in the manuscript. I add the year-by-year age uncertainty of the GRATAISS model to **Fig. 8B** and include the discussion above into section 4.4 and 5 of the manuscript.

• **Speleothem example**: In my opinion, the speleothem example is neither necessary nor constructive. As the model is currently written, it is not designed for speleothems (L167-170) and the speleothem example only serves to highlight this. In most cases, speleothems violate the assumption of sinusoidal oscillation in δ18O and thus would likely benefit from a different age modelling approach. The manuscript is already quite long with several examples, and there's no need to add length by explaining why such systems are difficult to model (e.g., L477-496; L588-605).

I disagree with the statement that the speleothem example is not constructive in the manuscript. The example serves to highlight the limitations of the ShellChron model while at the same time showing how the model performs when pushed towards these limitations. The speleothem example thereby serves as a benchmark for a worst-case scenario when applying ShellChron on troublesome or poorly constrained $\delta^{18}O_c$ records. I do agree that most speleothems violate the assumption of sinusoidal temperature or $\delta^{18}O_w$ oscillations (sinusoidal $\delta^{18}O_c$ oscillations are not assumed in the model, as the modelled $\delta^{18}O_c$ is built up of sinusoidal temperature and skewed sinusoidal growth rate records). However, sinusoidal oscillations in $\delta^{18}O_c$ of the drip water feeding speleothems have been demonstrated in cave monitoring studies (e.g. Baldini et al., 2008; Van Rampelbergh et al., 2014; Vansteenberge et al., 2019; see discussion in section 4.3 and 5). This caveat allows the assumptions of ShellChron to be discussed (as pointed out by the referee) and the modular character of ShellChron to be showcased with an outlook towards future improvements. In summary, I prefer to leave the speleothem example in the manuscript, unless the editor has a strong objection to this.

**Specific comments**
L217: add comma between "included" and "which"
Done
L358: change "all." to "al."
Done
L444-446: awkward phrasing; consider revising to "The lower sampling resolution later in the record mutes this variability and further illustrates that..."
This has been rephrased accordingly
L468: "real" feels colloquial and nonscientific; I'd recommend changing "real" to "known", both in the text and the figures (e.g., "the "known" age of the samples in these natural carbonates is not truly known.")
Good comment, I agree and rephrased to "known" throughout the manuscript and the figures

L476: remove "very" (such descriptors are unquantifiable)
Agreed, "very" was removed.
L671: change "eliminate" to "minimize"
Agreed, this has been rephrased
Figure 2: change the blue bar to reflect the window used in the Figure 4 example.
The blue bar has been moved to the first growth year in the revised version, in accordance with the window used in Fig. 4.
Figure 7: for consistency with Figure 6, I suggest keeping the isotope residuals histogram in the upper left corner and the date residuals histogram in the lower right corner
A good suggestion, I replaced the histograms accordingly in the revised Fig. 7

**New references cited**
Judd, Emily J., Linda C. Ivany, Robert M. DeConto, Anna Ruth W. Halberstadt, Nicole M. Miklus, Christopher K. Junium, and Benjamin T. Uveges (2019) "Seasonally resolved proxy data from the Antarctic Peninsula support a heterogeneous middle Eocene Southern Ocean." *Paleoceanography and Paleoclimatology* 34(5).
Tierney, Jessica E., Christopher J. Poulsen, Isabel P. Montanez, Tripti Bhattacharya, Ran Feng, Heather L. Ford, Barbel Honisch et al. (2020). "Past climates inform our future." *Science* 370(6517).

**Reviewer #3**
My comments here are informed by the original ShellChron manuscript, two sets of reviews, the revised manuscript, and detailed author responses to the original reviews, all of which I received in order to provide this feedback.

ShellChron is introduced as an open source, R-based package with which to interrogate subannually resolved proxy data from records with nominally sinusoidal input variables. The package builds upon a series of earlier publications by other authors over two+ decades detailing computational approaches with which to recover seasonally resolved environmental parameters from proxy data archived in accretionary, mineralized, largely biogenic records. A series of model and real test cases are explored so as to evaluate the performance of this model with data sets of varying internal complexity. Advantages and limitations are discussed in some detail, along with the range of possible applications.

I found the first round of reviews of this manuscript to be thorough and detailed. They identified several specific shortcomings of the initial iteration of the model that led to spurious or systematically biased output, enabling the author to refine and improve the code. De Winter's responses to reviewer feedback, as evidenced both in his written text and in the new rounds of model performance, are substantive, thoughtful, and thorough. Most important was the tendency of the model to underestimate observed seasonal range in the carbonate d18O values of test cases, a function in part of too-broad constraints on how much growth varies over a year (a problem identified also by Judd et al) and in how time was equated to distance along the accretionary axis. Careful examination of and adjustments to the code identified the sources of this problem and new simulations demonstrate that the issue appears to be well resolved. Test cases illustrate very good agreement between data and model. This is a nice package!

The major new advantage of ShellChron is that it combines the continuous moving window approach of earlier models with the intra-annually varying growth rate parameterized by the recent Judd et al. 2018 model for individual years of data and adds an approach for propagating error in both the X and Y axes through to the model output to generate a confidence envelope. As well, ShellChron expands availability of the Judd et al. approach from the original MATLAB to the R language, and its modular design allows for the user to parameterize a function describing variable d18O water and/or to use a seasonal growth model other than a skewed sinusoid. This, combined with the detailed and clear manuscript, all makes the model quite accessible to a wide audience and applicable to a range of situations. Since it incorporates and builds from previous work though, I would urge him to explicitly recommend citing both this paper/program and the Judd paper together. That way both the R package and the work that inspired it can receive adequate credit. Judd is an early career researcher, and we know how important citation rates are at that crucial career stage.

I would like to thank Referee #3 for his kind review of my revised manuscript and model, and for the suggestion. I fully agree that, while ShellChron is in many ways different from the GRATAISS model,

the principles underlying it are very much derived from the model by Judd et al. I will therefore include the suggestion to cite this important work alongside this manuscript whenever ShellChron is used by the community. I added this suggestion at the end of the abstract where it will be most visible, but if the editor believes there is a more appropriate place for this statement, I would be happy to move it elsewhere.

Minor questions/edits:
Lines 258-260 (and elsewhere) – should this be 'differential weighting', rather than 'weighing' in this discussion?
Agreed, I rephrased "weighing" to "weighting" throughout the manuscript
Why in Figure 4A and 4B are full sinusoids not depicted?
The full model result (i.e. the age estimate for each of the datapoints in the example window) is depicted in these figures. See my reply to a comment by Referee #1 for an explanation of why the difference between the age of the first and last datapoint in the window is less than 365 years.
Line 339 – some text missing?
No text is missing here. Instead, the sentence starting with "in which…" explains the meaning of the symbols in the above formula 10.
Line 358 – should be 'et al.'
This is rephrased accordingly
Line 418 – 'with in'
Line 418 already contains the phrasing "with in". It is not clear to me if the reviewer suggests any rephrasing is needed here.
Figure 6C and F need a bit more description and qualification to make them clear. As well, variation in 6C looks to be (nearly) significant over the year, but this is minimized in the text. Worth more discussion?
I added more detailed description of Fig. 6C and F in the caption. The age offsets in these figures are indeed significant in some monthly time bins. These instances are now discussed in the manuscript text. See also my reply to comments by Referee #1 on these figures.
Line 566 – 'within' should be 'with'
Agreed, this is rephrased accordingly

**References**

Baldini, J. U. L., McDermott, F., Hoffmann, D. L., Richards, D. A., and Clipson, N.: Very high-frequency and seasonal cave atmosphere PCO2 variability: Implications for stalagmite growth and oxygen isotope-based paleoclimate records, Earth and Planetary Science Letters, 272, 118–129, https://doi.org/10.1016/j.epsl.2008.04.031, 2008.

Van Rampelbergh, M., Verheyden, S., Allan, M., Quinif, Y., Keppens, E., and Claeys, P.: Seasonal variations recorded in cave monitoring results and a 10 year monthly resolved speleothem δ18O and δ13C record from the Han-sur-Lesse cave, Belgium, 10, 1821–1856, 2014.

Vansteenberge, S., Winter, N. de, Sinnesael, M., Verheyden, S., Goderis, S., Malderen, S. J. M. V., Vanhaecke, F., and Claeys, P.: Reconstructing seasonality through stable isotope and trace element analysis of the Proserpine stalagmite, Han-sur-Lesse Cave, Belgium: indications for climate-driven changes during the last 400 years, 1–32, https://doi.org/10.5194/cp-2019-78, 2019.

Ivany, L. C. and Judd, E. J.: Deciphering Temperature Seasonality in Earth's Ancient Oceans, 50, 123-152, https://doi.org/10.1146/annurev-earth-032320-095156, 2022.